# Scalable Supervising Software Agents with Patch Reasoner

## Abstract

While large language model agents have advanced software engineering tasks, the unscalable nature of existing test-based supervision is limiting the potential improvement of data scaling. The reason is twofold: (1) building and running test sandbox is rather heavy and fragile, and (2) data with high-coverage tests is naturally rare and threatened by test hacking via edge cases. In this paper, we propose R4P, a patch verifier model to provide scalable rewards for training and testing SWE agents via reasoning. We consider that patch verification is fundamentally a reasoning task, mirroring how human repository maintainers review patches without writing and running new reproduction tests. To obtain sufficient reference and reduce the risk of reward hacking, R4P uses a group-wise objective for RL training, enabling it to verify multiple patches against each other's modification and gain a dense reward for stable training. R4P achieves 72.2% Acc. for verifying patches from SWE-bench-verified, surpassing OpenAI o3. To demonstrate R4P's practicality, we design and train a lite scaffold, Mini-SE, with *pure* reinforcement learning where all rewards are derived from R4P. As a result, Mini-SE achieves 26.2% Pass@1 on SWE-bench-verified, showing a 10.0% improvement over the original Qwen3-32B. This can be further improved to 33.8% with R4P for test-time scaling. The stable scaling curves in both RL test rewards and test-time accuracy reflect R4P's practical utility for scalable supervision on software agents.

## 1 Introduction

Large language models (LLMs) agents have made notable progress in software engineering (SWE) thanks to rule-based reinforcement learning (RL) and test-time scaling (TTS) (Jimenez et al., 2023; Luo et al., 2025a). However, unlike tasks with extensive publicly available and easy-to-check answers (e.g., math), SWE tasks naturally have *non-unique* answers (patches) that are hard to *formally verify* (ter Beek et al., 2024). Thus, *testing* was adopted as a proxy approach (Tihanyi et al., 2025) for verifying patches in SWE's RL and TTS. Yet, testing is *heavy*, hindering data scaling in SWE. Specifically, building test environment (e.g., Docker images) for each data instance is highly labor-intensive (Pan et al., 2024; Jain et al., 2025), and maintaining sandbox instances (e.g., Docker containers) for parallel testing is unstable due to surge workload (Luo et al., 2025a). Furthermore, the inherent *test coverage problem* (Yu et al., 2025) also harms the reliability of test results, making tests being easily *hacked* by edge cases when expanding data to less maintained GitHub projects. We found that only 28.11% of resolved issues have corresponding tests on projects with over 500 stars (Appendix A). This hinders the utilization of vast, unlabeled data from the open-source community.

We identify the core challenge is *how to acquire test-free supervision for patches at scale*. Ideally, a reward model (Minghao Yang, 2024; Shiwen et al., 2024; Chen et al., 2025) could serve as an alternative patch verifier. However, they are trained to judge the *relative* preference of solutions than their *absolute* correctness. Since SWE is a hard reasoning task, all LLM rollout answers may be incorrect and should not be rewarded, making such relative assessment inadequate. Furthermore, without test execution, for an LLM to reliably identify subtle incompatibilities, it needs an exhaustive, agentic search across the repository's call graph and dependencies. However, this approach makes the verification process as hard and slow as the patch generation process itself (Meng et al., 2024), making it impractical for scalable supervision. To mitigate this, some test-time verifiers (Pan et al., 2024; Jain et al., 2025; Luo et al., 2025a) use certain agent's trajectories as an additional input, trained to

maximize the log probability of token YES or NO from a process-oriented perspective. However, such task formulation creates a strong coupling between the model and a specific interaction style, hindering its ability to generalize to other agents.

In this paper, we introduce *R4P*, a *reasoning* patch verifier model for SWE agents that do not rely on any golden test, developer patch, agent trajectory, or run-time sandbox. It enables efficient data scaling for real-world issues that are untested or have low test coverage, offering potential for continual learning beyond the constraints of test-based sandbox data. We consider patch verification is fundamentally a reasoning task which can be improved via RL. This mirrors how real-world repository maintainers evaluate pull request (PR) patches based on their understanding of the repository and a detailed analysis of the patch, rather than relying on writing and running test scripts to judge correctness (Baum et al., 2016). To avoid reward hacking and enable effective test-free verification, R4P adopts a group-wise training objective. Specifically, for a given issue and a set of candidate patches, R4P assesses each patch by comparing it against others in the group for mutual contextual information, compensating for the absence of tests and facilitating the detection of subtle errors. Moreover, by expanding the verifier's solution space beyond binary classification, R4P reduces the risk of reward hacking and offers a denser reward compared to the original sparse bipolar reward. This leads to more stable training and improved convergence. We evaluate R4P on patch generated by four different agents from SWE-bench-verified Experiments. When built upon Qwen2.5-Coder-32B-Instruct, R4P achieves 72.2% accuracy, surpassing advanced models like OpenAI o3.

To further demonstrate R4P's practicality, we developed *Mini-SE*, a lite, execution-free agentic scaffold with issue-resolving-oriented code *search* and *edit* capabilities. We then use R4P for supervision and train Mini-SE based on Qwen3-32B model on R2E-Gym issues without using its sandbox. While Mini-SE does not test its generated patches during rollout, its verification burden is transferred to R4P for patch selection for test-time scaling. The Pass@1 resolution rate on SWE-bench-verified steadily improves with more training data and finally reaches 26.2%, outperforming Lingma Agent + Lingma SWE-GPT-72B (Ma et al., 2024). With R4P for patch selection, it can be further improved to 33.8%. In addition, R4P verifies each patch within a second on average, much faster than the minute-level time cost of testing. These illustrates the practicality of reasoning-based verification.

## 2 PRELIMINARIES

**Reinforcement Learning**: Reinforcement learning (RL) aims to learn an optimal policy $\pi_\theta$ that maximizes the expected cumulative discounted reward $r$ when interacting with an environment. In the context of auto-regressive LLM, state at step $t$ is the concatenation of prompt $x$ and current response $y_{<t}$, and the action is the t-th token $y_t$. Emprically, policy gradient methods are widely used to directly optimizes this objective $J$ and updates model parameters $\theta$:

$$\nabla_\theta J(\theta) = \mathbb{E}_{x \sim D, y \sim \pi_\theta} \left[ \sum_{t=0}^{T} \nabla_\theta \log \pi_\theta(y_t|x, y_{<t}) A_t \right] \quad (1)$$

For scalable rewarding, reward models are trained to provide a dense, normalized scalar reward (e.g., $r_s \in [0, 1)$), reflecting human preference rankings of answers. They have shown strength in human-AI alignment tasks like writing and safety. More recently, sparse verifiable rewards (e.g., $r_v = \{0, 1\}$) are proven to be effective on reasoning tasks with ground-truth correctness like math (Shao et al., 2024) and coding (Luo et al., 2025b), giving rise to a new domain, Reinforcement Learning with Verifiable Rewards (RLVR) (Schulman et al., 2017; Guo et al., 2025).

**Test-Time Scaling**: Test-time scaling (TTS) refers to approaches of utilizing test-time compute to enhance the answer quality. Given a certain test-time compute budget $N$, TTS aims to maximize the expectation $E$ of the number of model output $y^*(x)$ equals to the ground-truth $y$, which is notated as indicator function $\mathbb{I}_{y=y^*(x)}$ for a given prompt $x$, which equals 1 if the predicted label $y_i$ for patch $p_i$ matches the ground-truth label $y^*(p_i)$, and 0 otherwise (Snell et al., 2024). Assume the output tokens $y$ follows the model distribution $D(\theta|N, x)$, an ideal strategy $\theta^*$ shoulde be:

$$\theta^*(N) = \text{argmax}_\theta \left( \mathbb{E}_{y \sim D(\theta|N,x)} \left[ \mathbb{I}_{y=y^*(x)} \right] \right) \quad (2)$$

Common TTS approches include expanding multi-turn searching or revisions during rollout (Madaan et al., 2023; Wang et al., 2024) and sampling multiple candidates for filtering or selec-

tion after rollout (Wang et al., 2022; Xia et al., 2024) against verifier. This verifier could also be rules or reward models, as they requires similar supervision signals like rewards in RL. For example, Snell et al. train a process reward model to enable Monte Carlo tree search (Sutton et al., 1998) during inference, while Xia et al. use an LLM for scaling tests to filter flawed answers after rollout.

**LLM for Issue Resolving**: Software Enginnering (SWE) tasks have increasingly gained attention, especially for real-world repository-level issue resolving tasks (Jimenez et al., 2023). Given an issue description in natural language (e.g., a feature request or a bug report), the models need to scan the whole repositroy to understand the problem, locate the place for editing, and finally submit a patch as the answer. The golden tests (i.e., the unit tests submitted in the developer patches) and corresponding sanbox environments (typically based on Docker) will then be applied to verify the patch. While such test-based verification is effective for benchmarking, it cannot provide scalable supervision for improving model's SWE capability. We identified two major problems:

*P.1: Testing is heavy*: Even with the help of LLM, building a single issue's Docker image takes about 7 human minutes (Jain et al., 2025). Thus, existing datasets contain less than 5,000 real-world executable issue instances (Pan et al., 2024). While *issue synthesis* allows for data scaling in a reusable sandbox via commit backtranslation (Yang et al., 2025), they are proven to be much less effective than real-world issues (Luo et al., 2025a). Furthermore, maintaining hundreds of Docker containers (e.g., 64 batch size $\times$ 8 rollouts) for parallel online rewarding is highly unstable, as containers are prone to crashing under heavy peer load (Luo et al., 2025a). In addition, while a timeout threshold is necessary to avoid process blocking from infinite loop of flawed patch, it can also introduce false negative rewards on instances running heavy test suites.

*P.2: Test coverage problem*: In practice, developer's unit tests often fail to cover all edge cases, allowing tricky patches to pass without genuinely solving the issue (Mockus et al., 2009). This problem persists even in widely recognized high quality dataset, SWE-bench (Yu et al., 2025), and will be eventually more exacerbated in less actively maintained repositories for RL scaling. Such test coverage problem will unavoidable results in reward hacking after running out of existing real-world high quality issues. Thus, it is challenging to continue exploiting existing tests for data scaling in-the-wild. Furthermore, our study shows that 28.11% issues from projects with more than 500 stars even do not contain *any* tests, let along the test coverage problems (see Appendix A). This prevents the exploitation of vast, unlabeled data from the open-source community (e.g., GitHub, Jira).

**Challenges of Model-based Verification**: A way to scale supervision is to train a reward model (RM) as patch verifier, which judges LLM responses without relying on tests and environments. However, it is usually infeasible in practice to incorporate RMs into SWE due to two challenges:

*C.1 Lack of sufficient reference*: In addition to scalar and trajectory RMs discussed in Sec. 1, we explored the capability of advanced LLMs as RMs for patch verification (see Sec. 5.3). Our analysis of failure cases reveals the reason is due to the lack of dependency context for the patch, preventing the model from identifying subtle incompatibilities or bugs with the existing codebase. While agentic search is a potential solution, precisely identifying all issue-relevant context is both difficult and inefficient, as an issue in a large project can be far-reaching (Meng et al., 2024).

*C.2 Sparsity of outcome space*: Since the output space of patch verification is binary (Pass/Fail), the supervision signal for RM training is very sparses, making naively mapping input issue-patch pairs to a Pass/Fail label very challenging. In practice, diverse patches can solve the same issue without a dense, relative relationship between each other, which further precludes the use of dense training objectives like Bradley-Terry model (Kendall & Smith, 1940). In addition, simple binary classification SFT offers limited benefits to open-source models. Furthermore, the binary outcome reward is very easy to hack, making the training unstable.

## 3 R4P

To provide scalable patch verification for SWE tasks, we propose R4P, a reasoning reward model for patch verification trained via pure rule-based RL. We consider patch verification is naturally a reasoning task, as human developers review pull requests (PRs) by reasoning about the static code changes, rather than designing new, issue-specific golden tests for every submission. R4P enables efficient data scaling for unlabeled or low-test-coverage real-world issues, thereby supporting continuous learning beyond the limited test-based sandbox data.

**Task formulation** To overcome the challenges of reference deficiency (*C.1*) and sparse rewards (*C.2*), R4P introduces a group-wise task formulation. Given a SWE issue $I$ in issue description and a group of patches $P = [p_1, p_2, ..., p_N]$ trying to resolve the issue, R4P is expected to generate a sequence of tokens, which include the reasoning tokens and the ID of the correct patches $y_i$:

---

You are a software expert. You will be given a software issue and some patch candidates in user query. You need to judge which patch(es) can resolve the issue. Carefully review, critic, and compare the given candidates. You need to first think about the reasoning process in the mind until you get the final answer. Finally, put the ID(s) of correct patch candidates in \boxed{}.

```
[ISSUE] {user issue I} [/ISSUE]
[PATCH 1] {agent patch P_1} [/PATCH 1]
...
[PATCH N] {agent patch P_N} [/PATCH N]
```

---

This group-level verification fully leverages the non-unique and diverse characteristics of patches, since the models are tended to modify various positions with different code changes, where each patch provides sufficient information for judging other patches. As a result, R4P could cross-reference all patches' edit locations and content, inferring potential context and identifying subtle errors in one patch by observing others. Furthermore, this group-wise approach transforms the binary outcome space into a much denser one (e.g., $\sum C_N^i$ possibilities for random selecting correct patches from a group $P = [p_1, ..., p_N]$), which provides a richer supervision signal and significantly mitigates the reward hacking risk inherent in simple binary classification tasks. Specifically, during RL training, the reward of R4P $r(P)$ is calculated by the indicator function $\mathbb{I}_{y_i=y^*(p_i)}$, which equals 1 if the predicted label $y_i$ for patch $p_i$ matches the ground-truth label $y^*(p_i)$, and 0 otherwise.

$$r(P) = \begin{cases} \frac{1}{N} \sum_{i=1}^N \mathbb{I}_{y_i=y^*(p_i)} & \text{if answer is boxed} \\ 0 & \text{if answe is unboxed} \end{cases} \tag{3}$$

**Reward modeling** To avoid reward hacking caused by the model's random guesses on a patch's binary correctness when uncertain, we adopt a continuous group-wise reward requiring the model to verify as many correct patches as possible (i.e., group-wise accuracy). The intuition is: a reasoning that correctly verifies more patches should be considered better than one that verifies fewer, even though certain random guessing is inevitable. Thus, the reward clearly reflect how much better one reasoning result is compared to another, rather than treating real correct reasoning as equivalent to a randomly guessed correct answer. Note that when the number of patches to be verified ($N_v$) exceeds the group size ($N_t$), the set of patches is partitioned into smaller groups for verification, with the total number of inference calls being $\lceil N_v/N_t \rceil$. Conversely, if the number of patches is less than $N_t$, the group is padded with empty patches to match the input format used during training, ensuring consistent model behavior.

**Data sampling** To collect a dataset of patches that closely resemble those generated by contemporary SWE agents, we use Claude-3.7-sonnet on OpenHands (Wang et al., 2024) scaffold to resolve issues in SWE-Gym (Pan et al., 2024). For each of the 2,438 issue instances in SWE-Gym, we sampled 6 patch candidates, resulting in a total of 14,628 verified patches. Since the patch accuracy of this process is only 30%, the original dataset was highly imbalanced with a majority of incorrect patches. Thus, we filtered a subset of the incorrect patches to create a more balanced label distribution. These data is then used for RL training of R4P via GRPO (Guo et al., 2025) (Appendix B.1).

## 4 MINI-SE

To validate R4P's practicality, we aim to train an agent and evaluate its performance with all supervision from R4P. To achieve it, a straightforward approach is to adapt existing agent scaffolds (Yang et al., 2024; Wang et al., 2024). However, this faces two major challenges. First, their tool designs are often overly general. The bash-like or python-like commands often leads to excessively long agent trajectories due to atomic actions like `cd` or `ls`. Furthermore, some complex actions like

grep may have a vast argument space, which is error-prone for LLMs, involving multiple inter-actions of trial-and-errors. This makes the RL rollout and backward processes very inefficient for validating R4P's RL supervision capability. Second, they typically follow a "generate-then-verify" workflow, attempting to generate and execute tests to validate their own patches during rollout. However, since existing methods typically adopts a separate and more powerful TTS verifier for fi-nal patch selection (Jain et al., 2025), this in-loop self-verification introduces redundant interactions and further hampers training efficiency.

To enable an efficient validation of R4P's practicality, we introduce Mini-SE, a lightweight scaffold that focuses on issue resolving without overlong trial-and-error iterations and avoid dependency on executable environment. It contains:

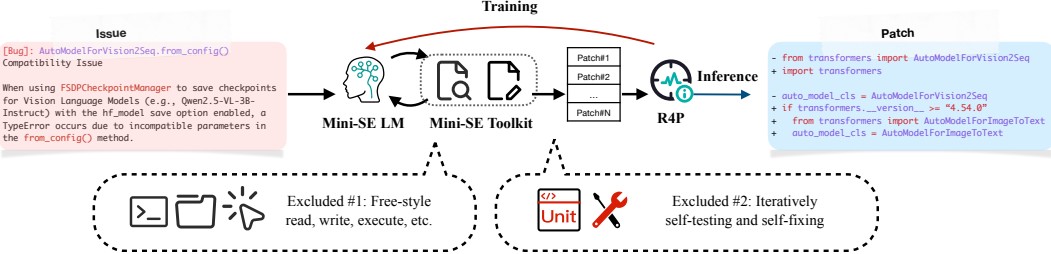

Figure 1: Mini-SE adopts an test-free issue-resolution-oriented tool design strategy: **Search**: input an *entity* name, output a file path and *code* snippet. **Edit**: input a file path, old code and new code snippet, output a diff *patch*. This design prevents the inefficient, redundant iterations caused by free-style exploration during training rollouts. It also removes self-testing and fixing when generating patches since R4P could take this role for penalizing wrong ones during training and selecting the correct one during inference. This further improves efficiency and avoid relying on sandbox.

**Code Search**   A tool that maps an input entity's simple name to corresponding source code and file path. Upon issue input, it initialize a code graph database for entire repository, extracting all *class*, *method*, and *function* entities with static analyzer. To reduce the interaction turns, this tool returns the source code of *all* entities with that simple name (e.g., func). This *fuzzy matching* also reduces tool call failures due to incorrectly spelled fully qualified names (e.g., file.Class.func).

**Code Edit**   A tool that replace an old string to a new string in the input file path, which could work as *insert*, *delete*, and *modify* actions. Upon issue input, it checkouts the specific commit and creates a shallow clone of the target repository as a workspace. All modifications are validated using *syntax checker*, and any change with syntax errors is *discarded*. This check reduces failures from the LLM repeatedly trying to fix syntax errors introduced in previous wrong fixes.

**Rollout process of Mini-SE**   Mini-SE focuses solely on issue resolution, generally following a targeted *"entity → code → patch"* paradigm. Specifically, it begins with "symptom" entities men-tioned in the issue description, examines its code, and iteratively traces through the dependency chain to identify the "root cause" entities. Once located, it modifies the corresponding code and submits a patch. The absence of trial-and-error interactions due to incorrectly using general-purpose tools enables Mini-SE swiftly generating a large volume of candidate patches. In addition, Mini-SE decouples the verification process from the generation process. During rollout, it directly leverages the reward signal provided by R4P to reinforce correct patches and penalize incorrect ones with-out self-test-self-fix, leading to highly efficient training. The saved token budget can be reallocated to patch verifier during test-time patch selection. All of these features makes Mini-SE an efficient scaffold for validating R4P's practicality.

## 5 EXPERIMENT

### 5.1 EXPERIMENTAL SETUP

**Implementation**   We implement R4P using Qwen-2.5-Coder-Instruct-32B with group size N = 4, as it offers a balanced trade-off between patch verification capability and computational

Table 1: Comparison with general models. ("∗" mark means point-wise patch verification)

| Model | Acc | F1 | EM |
|---|---|---|---|
| claude-3.7-sonnet | 68.1 | 50.5 | 26.8 |
| claude-4-sonnet | 68.4 | 50.0 | 25.7 |
| gemini-2.5-pro | **72.7** | 56.6 | 34.6 |
| o4-mini | 68.5 | 52.0 | 29.6 |
| gpt-4o | 61.2 | 43.0 | 17.3 |
| o3 | 71.5 | 57.4 | 36.4 |
| qwen-2.5-coder-7b* | 54.0 | N/A | N/A |
| qwen-2.5-coder-7b | 60.0 | 43.3 | 17.9 |
| qwen-2.5-coder-32b* | 55.9 | N/A | N/A |
| qwen-2.5-coder-32b | 61.9 | 44.5 | 18.5 |
| R4P | 72.2 | **63.3** | **41.8** |

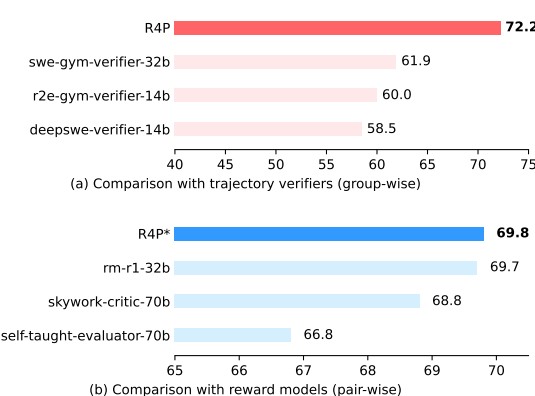

Figure 2: Comparision with specialized models. ("∗" mark means pair-wise patch selection)

cost. We do not adopt Qwen-3 series for R4P because enabling its thinking mode results in excessively long reasoning chains (around 32K tokens), which significantly increases computational overhead during both training and inference. Disabling thinking mode, however, leads to substantial performance degradation. In contrast, for Mini-SE, we base it on Qwen-3-32B because the agentic interaction pattern involves large volumes of tool-returned tokens rather than model-generated ones, making its training more efficient than that of R4P. Additionally, it demonstrates better tool-use instruction compliance compared to Qwen 2.5 series. Mini-SE's tarining details are in Appendix B.2.

**Datasets**    We collect patches generated by four different agents and LLMs from submissions in the SWE-bench-verified experiments, with a resolution rate of around 50% to ensure label balance (Appendix B.3). To ensure a fair comparison, we adopt the same group-wise formulation for general models as used in R4P. To maintain consistent task difficulty, all empty patches are removed. The final dataset comprises 1,340 patches, forming 335 group-wise instances. To further assess the practicality of R4P, we use real-world issues from R2E-Gym along with rewards generated by R4P to train Mini-SE via RL. This dataset contains 4,578 issues from 10 repositories, where we exclude any issues that overlap with R4P's training data. We evaluate Mini-SE on SWE-bench-verified, which consists of 500 issues with sandbox testing.

## 5.2    MAIN RESULTS

**Effectiveness on patch verification**    As detailed in Table 1, we compare R4P with various advanced generative and reasoning LLMs. R4P achieves a verification accuracy of 72.2%, surpassing strong proprietary models like OpenAI o3, despite R4P having a significantly smaller parameter scale. Since group-wise verification is a multi-choice retrieval task, we also report the F1-score and Exact Match ratio (details of all evaluation metrics are in Appendix B.3.). On these metrics, R4P achieved 63.3% F1 and 41.8% EM, outperforming all baselines. We also compare R4P with trajectory verifiers and general reward models. For trajectory verifiers, we use their original prompts and apply controlled decoding to compare the probabilities of YES and NO for judging patch correctness. As shown in Fig. 2 (a), R4P largely outperforms them, as it provides a scaffold-ignorant verification, which is much more generalizable. For reward models, since they can only provide relative preference on answers, we construct a pair-wise subset from our evaluation data, where each instance contains one correct and one incorrect patch from the same issue. While this formulation is unoptimized for R4P, it still achieves a 69.8% accuracy, slightly outperforming the state-of-the-art, as shown in Fig. 2 (b). This result underscores that R4P's core verification capabilities are robust and can be effectively generalized to other patch evaluation scenarios.

**Practicality on RL**    To validate if R4P's supervision could support RL scaling, we trained Mini-SE via *pure RL*. Notably, unlike normal SWE agents, Mini-SE is lightweight and does not test its patches during rollout. Thus, it naturally achieves a lower Pass@1 score in the absence of an

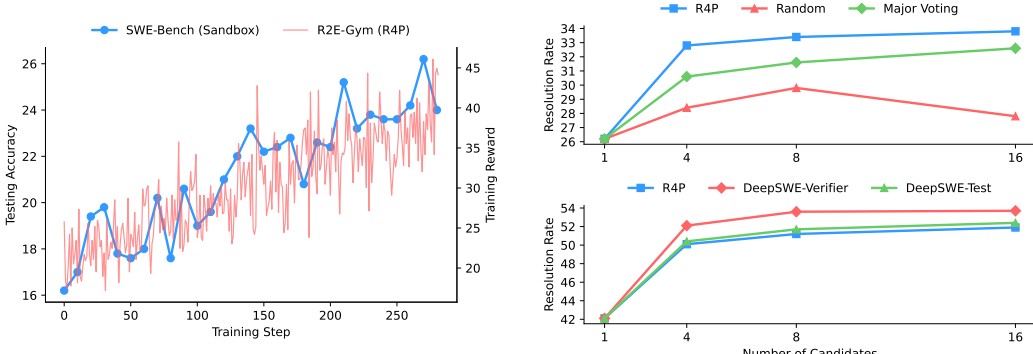

Figure 3: **Left:** Mini-SE's training rewards and testing accuracy in RL process. **Upper Right:** Mini-SE's resolution rates w.r.t. different TTS strategy. **Lower Right:** R4P's performance on selecting DeepSWE's patches. The scaling curves illustrates R4P's practicality and generalizability.

extra test-time patch verifier. Nevertheless, as shown in Fig. 3, Mini-SE achieved a 26.2% Pass@1 within 300 steps, boosting the base Qwen3-32B model performance by 10.0%. Furthermore, the test accuracy increases steadily with training rewards, indicating that R4P provides stable supervision signals without leading to reward collapse. This feature enables R4P to further supervise agents on vast, untested issues in open-source community, illustrating its practicality for agentic RL training.

**Practicality on TTS**  To validate R4P's effectiveness for TTS, we sample 16 patches from Mini-SE on the SWE-bench-verified. We employed a two-round process: the patches are grouped for R4P verification, and those predicted as incorrect patches will be filtered out. Then we prompt R4P to output the single most likely correct patch from all remaining candidates as output. As shown in Fig. 3, Mini-SE's resolution rate increases steadily with the number of candidate patches evaluated. When all 16 patches are considered, it reaches 33.8%. To further show its generalizability, we employ R4P on DeepSWE's patches for selection. While both of the test agent and verifier model of DeepSWE are optimized for its own trajectory, R4P achieves a comparable results of the test agent.

**Efficiency on patch verification**  Fig. 4 shows the distribution of the average time required to verify each patch per step when R4P is deployed on 2x80GB GPUs via vLLM. We compare it against the time taken to test the golden patches for 500 instances from swe-bench-verified using a CPU server with 64 cores and 128GB of RAM. The average time per instance for R4P does not exceed 1 second, significantly outperforming the average testing time per instance of 50 seconds. Furthermore, patches generated by LLMs often contain algorithmic inefficiencies or infinite loops. During the training of Mini-SE, we observed that 59.2% (29/49) of validation instances reached the 30-minute timeout of SWE-bench harness. This demonstrates the superior efficiency of R4P's group-wise reasoning-based verification compared to testing-based verification.

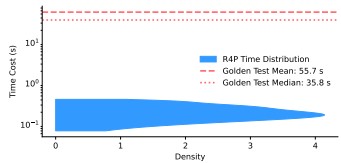

Figure 4: Time cost per patch.

## 5.3 ANALYSIS AND DISCUSSION

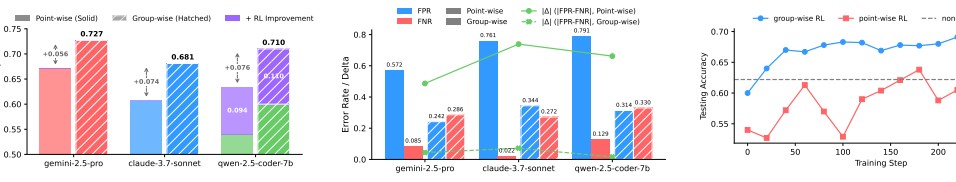

Figure 5: Task formulation          Figure 6: Bias of FPR/FNR          Figure 7: RL convergence

**Impact of group-wise patch reasoning.** Verifying a patch in isolation is a significant challenge for LLMs. As shown in Fig. 5, even advanced models like Gemini struggle with this task, while smaller models like Qwen-7B perform at chance level and do not improve much even after targeted RL. However, with group-wise formulation, the performance is improved across all models. With RL, it can be further improved by 10%.

We identify this improvement stems from two aspects. First, the group-wise reasoning mitigates the tendency of LLMs to *over-estimate* a patch's quality when evaluated in isolation. As shown in Fig. 6, point-wise formulation shows a high false-positive rate, confirming that models inherently struggle to identify errors without sufficient context. In contrast, group-wise formulation enables LLMs to cross-reference multiple candidate patches to identify the ignorable subtle bugs: (1) When multiple patches compete to modify adjacent code snippets, the correct one could be distinct against the incorrect ones (e.g., "`v=dict.value`" compares to "`v=dict.get(value,None)`"); (2) When they modify different code snippets, they provide mutual context that a single patch lacks. As shown in Fig. 8, accuracy approaches 90% when patches in a group are either very similar or very different, indicating that both conditions simplify the verification. Therefore, it mitigate the threat of over-estimate. Second, the group-wise formulation resolves the issue of unstable RL due to the sparse outcome space of point-wise verification. As Fig. 7 illustrates, group-wise formulation enables smooth convergence, where point-wise RL training is highly unstable, and its peak performance barely surpasses that of the end-to-end non-reasoning supervised fine-tuning (SFT).

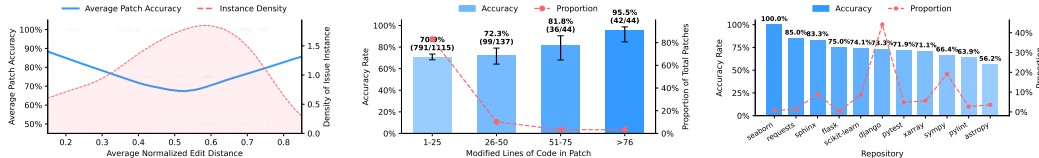

Figure 8: Distribution of accuracy on edit distance

Figure 9: Relations of accuracy and edited lines in patch

Figure 10: Accuracy distribution across repositories

**Application scope** We analyze R4P's application scope to show its comparative advantages and disadvantages. As shown in Fig. 9, R4P's verification accuracy generally appears to be positively correlated with the number of edited lines in a patch, as it cross-reference patches' edition content and position as context for effective patch verification. In addition, Fig. 10 demonstrates R4P's robust performance across different repositories, highlighting its generalizability. Furthermore, as discussed in Fig. 8, R4P's accuracy improves when patches within a group are either highly similar or highly dissimilar. Thus, we suggests several best practices for R4P application: (1) prioritize issues with shorter resolution periods (which is accessible via GitHub API), (2) sampling redundant rollouts and reject rewarding on extremely short patches, and (3) assemble patch groups with either very high/low mutual similarity rather than uniformly distributed for verification.

**Reward consistency** We analyzed the distribution of predictions that diverged from the majority vote. As shown in Fig. 11, when sampling 32 times with temperature=1.0, the results indicate high consistency: 88.8% of patches had no predictions deviating from the majority, and 3.2% had only one deviation across all runs. Overall prediction fluctuations is fewer than 8% of cases. Such stability ensures that our reasoning-based verification provides consistent rewards online, improving the reproducibility of supervised training and reducing the needs for inefficient majority voting on rewarding.

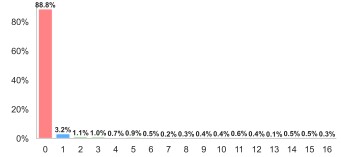

Figure 11: Deviation of majority

**Reward strategy between model-based reasoning and sandbox testing** The model-based reward is unavoidable contains noise. Thus, the use of model-based rewards should be based on the premise that *the benefits from data and supervision scaling will outweigh the losses from noisy signals*. Empirically, the stable reward curve in Fig. 3 shows that R4P's reward is overall stable and will not trigger reward collapse. Furthermore, we observe that the gradient norm of Mini-SE stabilizes around 0.01. Compared to existing RL-based agents (Luo et al., 2025a), Mini-SE's gradient norm is

approximately 25% higher. This suggests that although some noisy supervision creates conflicting gradients, the overall update direction remains correct. Given R4P's ability to scale data efficiently, this nosiy yet stable supervision is an acceptable trade-off.

# 6 RELATED WORK

**SWE data scaling** Data scaling has become a central challenge of SWE. Typically, people mines issues from GitHub and creating sandbox for testing (Pan et al., 2024), which is very labor-intensive. To facilitate it, one way is LLM-assisted sandbox construction (Hu et al., 2025; Yang et al., 2025). However, effort is still required for checking sandbox correctness (Hu et al., 2025). Another way is issue generation and commit backtranslation to mutate numerous instances from several seed instances (Jain et al., 2025; Yang et al., 2025). However, they often differ from real-world issues and empirically lead to suboptimal training performance (Luo et al., 2025a). A third approach relies on smilarity metrics to supervising SWE sub-tasks sparately Ma et al. (2025); Wei et al. (2025); Xie et al. (2025). However, since correct patches are often non-unique, it can incorrectly penalize novel yet valid solutions, introducing noise and reducing sample efficiency. In contrast to these data-centric methods, we focus on expanding supervision rather than data, as a large number of diverse untested issues remain under-exploited in the whole open-source community.

**Reward models and verifiers** Reward models are intially designed to predict an answer's relative quality (Minghao Yang, 2024; Chen et al., 2025) for AI-human preference alignment. With the emergence of Reinforcement Learning with Verifiable Rewards (RLVR) and Monte Carlo-based RL algorithms, e.g., GRPO (Guo et al., 2025; Shao et al., 2024), rule-based rewards have gained prominence in reasoning tasks where absolute correctness can be determined. In such settings, reward models are often repurposed as verifiers during test-time selection (Pan et al., 2024; Jain et al., 2025; Luo et al., 2025a). However, SWE tasks lack easily accessible high-quality rule-based rewards from testing. Thus, we explores using reward models with absolute patch verification capability to provide scalable, deterministic supervision.

# 7 LIMITATION AND CONCLUSION

**Limitation** *First*, R4P's weights remain fixed after training. As the agent's policy improves, the static reward model may become misaligned with true answer quality (Gao et al., 2023). In furture work, we plan to explore periodic calibration using issues with high-quality sandbox during extended RL training, where discrepancies between R4P's predictions and actual test results could guide weight updates. *Second*, as a patch verifier, R4P are designed to supervise patch generation process and cannot solely provide end-to-end supervision for complex agents that perform in-loop self-verification via test generation and execution (Gao et al., 2025; Luo et al., 2025a). Future work will explore mixed training strategies, e.g., scaling normal issues with R4P to train straightforward patch generation while leveraging challenging issues with high-quality test sandbox to train patch generation with in-loop verification. *Third*, R4P and Mini-SE are Python-centric. To generalize to other languages, it is required to collect patches from their datasets and adapt the code search tool to their static analyzer. *Fourth*, R4P's performance correlates with multiple factors like group diversity as discussed in Sec. 5.3. To mitigate it, it might be helpful to follow our best practice suggestions.

**Conclusion** This work explores a model-based approach to address the scalability bottleneck in supervising software engineering agents due to reliance on testing. It introduce R4P, a reasoning-based patch verifier. R4P utilizes group-wise training objective to effectively verify patches against each other and mitigate instability during training its patch reasoning capability. Empirical results demonstrate that R4P achieves 72.2% verification accuracy, outperforming strong reasoning models like OpenAI o3. We further showcased its practical value by training Mini-SE, a fully test-free agent that leverages R4P for both reinforcement learning and test-time scaling. Mini-SE achieves 26.2% Pass@1 and 33.8% Best@16 on SWE-bench-verified, showing that reasoning-based supervision is a viable and powerful alternative to traditional test suites.

ETHICS STATEMENT

This work adheres to the ICLR Code of Ethics. It uses only publicly available datasets and involves no human subjects or sensitive personal data. We considered fairness, privacy, and security and found no ethical concerns; the authors take full responsibility.

REPRODUCIBILITY STATEMENT

All datasets are public, and preprocessing details appear in the Appendix. We release anonymized code and scripts—with configs, hyperparameters, and evaluation protocols—to fully reproduce our experiments and results in `https://anonymous.4open.science/r/pacth-verifier-4F53/README.md`.

LLM USE DISCLOSURE

LLM was used only for language polishing; it did not assist with the research content, and the authors are fully responsible for the manuscript.

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

## A Preliminary Study of Tested Issues

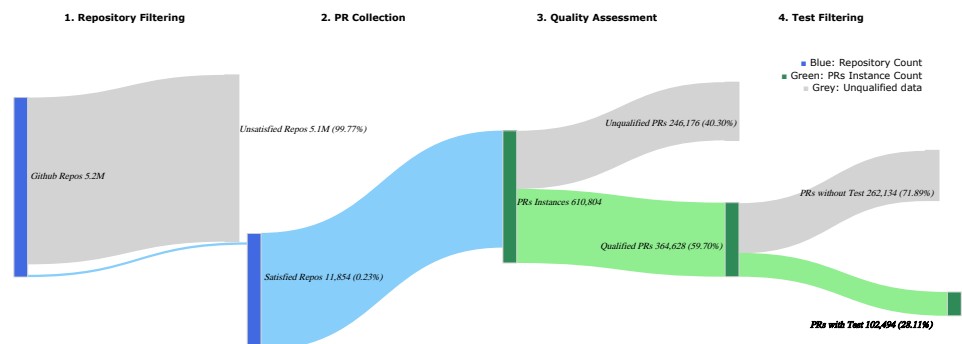

Figure 12: Visualization of data flow across four hierarchical steps in our GitHub issue curation pipeline: (1) *Repository Filtering*, (2) *PRs Collection*, (3) *Quality Assessment*, and (4) *Test Filtering*. The text in **Blue** indicates repository counts, the text in **Green** indicates issue–instance counts, and Grey flows denote data filtered out as unsatisfied or unqualified at each stage. Numbers and percentages shown in the diagram are computed within each step.

This study measures the prevalence of associated tests for issues and pull requests in open-source communities. We begin from the universe of 5.2 M public GitHub repositories and pre-filter to Python projects with at least 500 stars, excluding any repository already included in *SWE-Bench Verified*. The repository filtering leaves 11,854 eligible projects (0.23%). From these repositories, we harvest 610,804 PRs. We then apply a series of quality filters to the candidate issues: (i) the problem description must fall within 100–4,000 characters to align with the length distribution observed in *SWE-Bench Verified*; (ii) descriptions containing non-ASCII content (e.g., Chinese characters) are discarded; (iii) associated fixes must not modify non-Python files; (iv) patches must be small and self-contained, touching 1–5 files; and (v) issues embedding images in the description are removed to avoid non-textual ambiguity. Finally, in qualified PRs, we filter the patch without corresponding test changes, leaving only 102,494 qualified PRs with tests. As summarized in Fig. 12, this pipeline yields only **28.11%** PRs have corresponding tests.

## B Experiment Details

### B.1 Training R4P

**Training settings** We adopt GRPO algorithm for training R4P. Based on the insights from recent DAPO report, we set the clip ratio high to 0.28 and do not employ a KL loss in objective function. With a batch size of 128, we trained Qwen-2.5-Coder-Instruct fro 225 steps. To accelerate the convergence process, R4P generates 16 samples and performs 16 gradient backward in each step. Throughout training, the temperature is maintained at 1.0 and the learning rate is fixed at 1e-6. To implement the training, we adopt VeRL framework with vLLM engine for rollout.

**Dataset construction** As mentioned in Sec. 3, we sampled 6 patches for each issue in the SWE-gym sandbox. During training data construction, the group size was set to 4. To improve the sample efficiency of the patches, we created multiple instances for each issue by forming different combinations of patches, i.e., constructing $C_6^4$ combinations for each issue. However, since the average solve rate of Claude-3.7 with OpenHands on SWE-gym is only about 30%, directly use such label imbalanced data will lead to significant under-estimate tendency of LLM. Thus, we balanced the combinations such that each group contained a roughly equal number of instances with correct patches ranging from 0 to 4—each category accounting for approximately 50% of the total instances. Ultimately, this process yielded 7,599 training samples with a group size of 4.

**Prompt template** We provide the system and user prompt in our R4P implementation, where the verification group size N=4:

*System prompt*

```
You are a software expert. You will be given a software issue and some
    patch candidates in user query. You need to judge which patch(es) can
     resolve the issue. Carefully review, critic, and compare the given
    candidates. You need to first think about the reasoning process in
    the mind until you get the final answer. Finally, put the ID(s) of
    correct patch candidates within \boxed{}, e.g., \boxed{1}, \boxed{2,
    4}, \boxed{1, 2, 3, 4} (all correct), \boxed{} (all wrong).
```

*User prompt*

```
<issue>
{problem_statement}
</issue>
<patch-1>
{generated_patch_1}
<patch-1>
<patch-2>
{generated_patch_2}
<patch-2>
<patch-3>
{generated_patch_3}
<patch-3>
<patch-4>
{generated_patch_4}
<patch-4>
```

### B.2 Training Mini-SE

**Training settings**  We mainly follow the best practice provided by DeepSWE, including using leave-one-out and remove reward standard deviation for advantage estimation, filter out overlong trajectory without loss calculation, and normalize policy loss by sequence length. During training, we adopt fully on-policy training, with batch size of 64 and sampling 8 rollout per step. We also fix the temperature at 1.0 and the learning rate at 1e-6. To implement agentic RL, we adopt VeRL's AgentLoop framework for training, and use vLLM for rollout. To accelerate the training process, we extract all instance's code graph via tree-sitter in advance, which can be directly used for code search tool. We also checkout the copy of repositories in a separate directory for models to edit. All the code changes will be recorded in the final patches generated by git diff. In addition, we set a maximum limit of 50 rounds and 28K tokens during training. We deploy R4P model on a vLLM server for providing verification. The reward for agent's RL training is vanilla rule-based reward (1 for "correct" patch, 0 for "incorrect" patch). We finish the training process before 300 stpes.

**Prompt template**  We provide the system prompt and tool configs of Mini-SE for reference. The user prompt is the SWE-bench or R2E-gym problem statement itself.

*System prompt*

```
You are an expert AI software engineering agent. Your primary goal is to
    resolve a GitHub issue given in the user message. Following this
    workflow methodically:

1. Understand the problem:
   - Thoroughly comprehend the issue description, identifying core
       components and expected behavior
   - Determine reproduction steps and failure conditions
2. Explore and Locate:
   - Use `search_tool` to explore the relevant files, entities, and test
       cases related to the bug report
   - Locate the exact root cause of the bug
3. Develop and Fix:
   - Develop minimal, targeted code modifications to address the root
       cause
   - Use `edit_tool` to apply surgical patch. Aim for minimal, clean
       changes
```

```
4. Review and Revise:
   - Review the original reproduction steps to ensure the fix
       effectiveness
   - Review the relevant regression tests to avoid introducing any new
       bugs
   - Iterate using `search_tool` for review and `edit_tool` for revise
       until you confirm no edge cases are overlooked
5. Submit the patch:
   - Call `patch_submission` tool to generate a unix diff patch and
       submit it to the user when confirming full resolution
   - Ensure the final patch is non-empty before finishing this
       conversation
   - All code changes persist throughout the conversation and will be
       included in the final patch
```

*Tool configs*

```
type: "function"
function:
  name: "edit_tool"
  description: "A file edit tool that replaces an old string of text
      with a new string.\nNotes:\n1. The `old_str` parameter must match
      a segment of the file's content **exactly**. Pay close attention
      to whitespace, indentation, and newlines.\nThe edit will fail if `
      old_str` is not found, or if it is found multiple times in the
      file. Ensure `old_str` is unique enough to target the specific
      code block.\n3. `edit_tool` **permanently modifies the actual
      repository** (changes persist to the final outcome)."
  parameters:
    type: "object"
    properties:
      path:
        type: "string"
        description: "Relative file path e.g. `dir/file.py`."
      old_str:
        type: "string"
        description: "The exact string/content to be replaced in the file.
            This must be a unique match within the entire file."
      new_str:
        type: "string"
        description: "The new string that will replace `old_str`. Use an
            empty string to perform a deletion."
    required: ["path", "old_str", "new_str"]

type: "function"
function:
  name: "search_tool"
  description: "A code graph tool to show the source code of a function/
      class/class_method by its name.\nNotes:\n1. This tool is designed
      for Python code; it cannot find entities in other languages.\n2.
      If the `entity` name is not unique across the repository, the
      source code for all matching items will be displayed.\n3. This
      tool operates on a static, **initial snapshot of the repository**.
       Changes made with `edit_tool` will **not** be visible in the
      search results."
  parameters:
    type: "object"
    properties:
      construct:
        type: "string"
        description: "Type of program construct to search for."
        enum: ["function", "class", "class_method"]
      entity:
        type: "string"
```

```
        description: "The simple name of the entity to find (e.g., `
            my_function`, `MyClass`, `my_class_method`)."
      required: ["construct", "entity"]

  type: "function"
  function:
    name: "patch_submission"
    description: "Automatically generate a diff patch for existing code
        changes and submit it to the user."
    parameters:
      type: "object"
      properties: {}
      required: []
```

Note that we also provide a patch submission tool to enable a final check of LLM before ending the interaction and the real patch submission process.

### B.3 EVALUATION DATA

**Data source** Our evalation data for patch verification task is sampled from SWE-bench verified experiments, which include all the patch submissions of the agents on the leaderboard. Specifically, our training data is from the following submissions: (1) 20250520_openhands_devstral_small, (2) 20241029_openhands-codeact-2.1-sonnet-20241022, (3) 20240728_sweagent_gpt4o, (4) 20240620_sweagent_claude3.5sonnet, as they have a overall moderate accuracy for balancing patch labels. Note that we remove the empty patches and patches that has meaningless modifications (e.g., only comments or tests are chagned) to avoid model shortcut through these low-level features. We thereby formulate a final datasets with 1,340 patches from 335 issues, where the overall accuracy is 49.93% (669/1340).

**Prompt for reward model** We follow the official prompt tempalte of these pair-wise reward models:

*System prompt*

```
Please act as an impartial judge and evaluate the quality of the
    responses provided by two AI assistants to the user question
    displayed below. You should choose the assistant that follows the
    user's instructions and answers the user's question better. Your
    evaluation should consider factors such as the helpfulness,
    relevance, accuracy, depth, creativity, and level of detail of
    their responses. Begin your evaluation by comparing the two
    responses and provide a short explanation. Avoid any position
    biases and ensure that the order in which the responses were
    presented does not influence your decision. Do not allow the
    length of the responses to influence your evaluation. Do not favor
     certain names of the assistants. Be as objective as possible.
    After providing your explanation, output your final verdict by
    strictly following this format: \\"[[A]]\\" if assistant A is
    better, \\"[[B]]\\" if assistant B is better.
```

*User prompt*

```
[User Question]
{issue}

"[The Start of Assistant A's Answer]
{patch_1}
[The End of Assistant A's Answer]

[The Start of Assistant B's Answer]
{patch_2}
[The End of Assistant B's Answer]
```

**Prompt for trajectory verifier** We follow the official prompt template of these verifiers:

*System prompt*

```
You are an expert judge evaluating AI assistant interactions. Your task
    is to determine if the assistant successfully resolved the user's
    request.

Key evaluation criteria:
1. Did the assistant complete the main task requested by the user?
2. Did the assistant handle all edge cases and requirements specified?
3. Were there any errors or issues in the final solution?
4. Did the assistant verify the solution works as intended?

Respond only with "<judgement>YES</judgement>" or "<judgement>NO</
    judgement>".
```

where the user prompt is specific agent's trajectory.

**Evaluation metrics** If a dataset has $N$ patches and $M$ unique issues (where each issue corresponds to $K$ patches), then these $N$ patches can be grouped into $M$ patch groups (each of size $K$, i.e., $N = M \times K$). Below are the detailed definitions:

- **Acc.:** Accuracy is a *patch-level* evaluation metric that quantifies the proportion of correctly verified patches out of the total number. Let $y_i$ be the ground-truth label for patch $p_i$, and $y^*(p_i)$ be the model's prediction of its correctness. Using the indicator function $\mathbb{I}$ (which evaluates to 1 if the condition is true, and 0 otherwise), the accuracy is calculated as:

$$Acc = \frac{\sum_{i=1}^{N} \mathbb{I}_{y_i = y^*(p_i)}}{N} \qquad (4)$$

- **F1** The F1 score is a *group-level* evaluation metric. In our implementation, we calculate it using the set-theoretic definition (i.e., the Dice coefficient). For a given group $i$:
  - Let $A_i$ be the set of truly correct patches.
  - Let $B_i$ be the set of patches predicted as correct by the model.

  The F1 score for group $i$ is calculated as:

$$F1_i = \frac{2|A_i \cap B_i|}{|A_i| + |B_i|} \qquad (5)$$

This metric serves to measure the similarity between the predicted set ($B_i$) and the ground-truth set ($A_i$). Furthermore, it can provide an alternative view of a *retrieval task* i.e., *the task of "retrieving the correct patches from a set of candidate patches."* In this context, the F1 score represents the harmonic mean of Precision ($P_i$) and Recall ($R_i$):

$$F1_i = \frac{2P_i R_i}{P_i + R_i} = \frac{2TP_i}{2TP_i + FP_i + FN_i} = \frac{2|A_i \cap B_i|}{|A_i| + |B_i|} \qquad (6)$$

Note that in our implementation, we calculate the overall average F1 score for each individual group:

$$F1 = \frac{\sum_{i=1}^{M} F1_i}{M} \qquad (7)$$

To handle the boundary conditions where one or both sets might be empty, we consider:
  - $F1_i = 1$ if both sets are empty, as the model correctly predicted that no patch is correct.
  - $F1_i = 0$ if only one set is empty, as it is a completely incorrect prediction.

- **EM** Exact Match is a *group-level* metric that assesses the model's ability to correctly verify all patches within an entire group simultaneously. It is formally defined as the proportion of patch groups where the predicted set of correct patches exactly matches the ground-truth set:

$$EM = \frac{\sum_{i=1}^{M} \mathbb{I}_{|A_i| = |B_i|}}{M} \qquad (8)$$

918
919

## C  CASE STUDY

920
921
922
923

In this section, we provide a case study of django_django-13810. We release R4P's reasoning trajectory as well as those of proprietary models with accessible reasoning trajectory (Claude-3.7-Sonnet, Claude-4.0-Sonnet) as a reference.

924
925
926
927

**Issue and patches**   This issue is about Django's *MiddlewareNotUsed* error. In the original code, the handler variable is updated via *self.adapt_method_mode()* before the middleware instantiation triggers the exception. Thus, even though the middleware is skipped, the handler remains permanently modified (incorrectly adapted for subsequent middleware), leading to a sync/async mismatch.

928
929

To solve this issue, each patch has its own way:

930
931
932

- **Patch 1 & 4** (incorrect): Attempt to initialize the middleware *before* adapting the handler. This fails because middleware constructors typically expect the handler to already be in the specific mode (sync/async) required.

933
934
935

- **Patch 2** (correct): Assigns the adapted handler to a temporary variable. The main handler is only updated in the *else* block of the *try/except* statement. This ensures the update only occurs when no exception was raised.

936
937
938

- **Patch 3** (correct): Also uses a temporary variable, but performs the assignment inside the *try* block immediately. Since execution halts immediately if an exception is raised, this also prevents the previous bug.

939
940
941

In this case, R4P correctly identifies Patches 2 and 3 as correct patch, yet incorrectly consider Patch 4 as well. In contrast, the two Claude models only correctly identify either Patch 2 or 3.

942
943

Below are the raw content of the issue and patches, as well as the reasoning trajectory and corresponding analysis of R4P, Claude-3.7-Sonnet, and Claude-4.0-Sonnet.

944
945
946
947
948
949
950
951
952
953
954
955
956
957
958
959
960
961
962
963
964
965
966
967
968
969
970
971

```
<issue>
MiddlewareNotUsed leaves undesired side effects when loading middleware
    in ASGI context
Description

I experienced strange issues when working with ASGI, django-debug-toolbar
     and my own small middleware. It was hard problem to debug, I
    uploaded an example project here: https://github.com/hbielenia/asgi-
    djangotoolbar-bug (the name is misleading - I initially thought it's
    a bug with django-debug-toolbar).
The SESSION_FILE_PATH setting is intentionally broken to cause a 500
    error. When starting the application and accessing /admin (any
    location really, but I wanted to leave it at a minimum and didn't add
     any views) it gives TypeError: object HttpResponse can't be used in
    'await' expression. Commenting out asgi_djangotoolbar_bug.middleware.
    DummyMiddleware fixes the issue (in that I receive a 500
    ImproperlyConfigured exception). I'm not sure about the overall role
    of django-debug-toolbar here - removing it causes Daphne to return a
    500 error page but without debug information and there's no traceback
     in console either. I decided to leave it since it helped me
    approximate the causes of issue.
I notice that in https://github.com/django/django/blob/3.1.4/django/core/
    handlers/base.py#L58 while MiddlewareNotUsed causes the loop to skip
    futher processing and go to next middleware, it does leave handler
    variable overwritten with output of self.adapt_method_mode(). On next
     pass, this handler is passed to next middleware instance,
    disregarding all the previous checks for (lack of) async support.
    This likely causes the middleware chain to be "poisoned" from this
    point onwards, resulting in last middleware in response cycle to
    return an HttpResponse as a synchronous middleware would, instead of
    coroutine that is expected.
This is probably avoided by adding async support to my middleware, but
    unless I'm missing something docs indicate it should work as it is.
    It is my intention that it's applied only on synchronous requests, so
```

```
     I didn't make it async compatible on purpose. If it's intentional in
     Django that every middleware needs to support async if the
   application is run as ASGI app, the documentation should probably
   state that clearly. Though it kinda defeats the purpose of having
   async_capable = False flag in the first place.
</issue>

<patch-1>
diff --git a/django/core/handlers/base.py b/django/core/handlers/base.py
--- a/django/core/handlers/base.py
+++ b/django/core/handlers/base.py
@@ -51,11 +51,12 @@ class BaseHandler:
                 middleware_is_async = middleware_can_async
             try:
                 # Adapt handler, if needed.
+ # Adapt handler, if needed.
+ mw_instance = middleware(handler)
                 handler = self.adapt_method_mode(
                     middleware_is_async, handler, handler_is_async,
                     debug=settings.DEBUG, name='middleware %s' %
                         middleware_path,
                 )
- mw_instance = middleware(handler)
             except MiddlewareNotUsed as exc:
                 if settings.DEBUG:
                     if str(exc):
</patch-1>
<patch-2>
diff --git a/django/core/handlers/base.py b/django/core/handlers/base.py
--- a/django/core/handlers/base.py
+++ b/django/core/handlers/base.py
@@ -51,11 +51,11 @@ class BaseHandler:
                 middleware_is_async = middleware_can_async
             try:
                 # Adapt handler, if needed.
- handler = self.adapt_method_mode(
+ adapted_handler = self.adapt_method_mode(
                     middleware_is_async, handler, handler_is_async,
                     debug=settings.DEBUG, name='middleware %s' %
                         middleware_path,
                 )
- mw_instance = middleware(handler)
+ mw_instance = middleware(adapted_handler)
             except MiddlewareNotUsed as exc:
                 if settings.DEBUG:
                     if str(exc):
@@ -63,6 +63,9 @@ class BaseHandler:
                     else:
                         logger.debug('MiddlewareNotUsed: %r', middleware_path)
                 continue
+ else:
+ # Only update handler if middleware was not skipped
+ handler = adapted_handler

             if mw_instance is None:
                 raise ImproperlyConfigured(
</patch-2>

<patch-3>
diff --git a/django/core/handlers/base.py b/django/core/handlers/base.py
--- a/django/core/handlers/base.py
+++ b/django/core/handlers/base.py
@@ -50,12 +50,14 @@ class BaseHandler:
             else:
                 middleware_is_async = middleware_can_async
```

```
            try:
-       # Adapt handler, if needed.
-       handler = self.adapt_method_mode(
+       # Only adapt handler if the middleware is used
+       adapted_handler = self.adapt_method_mode(
                    middleware_is_async, handler, handler_is_async,
                    debug=settings.DEBUG, name='middleware %s' %
                        middleware_path,
                )
-       mw_instance = middleware(handler)
+       mw_instance = middleware(adapted_handler)
+       # If we get here, the middleware will be used, so update the handler
+       handler = adapted_handler
            except MiddlewareNotUsed as exc:
                if settings.DEBUG:
                    if str(exc):
</patch-3>

<patch-4>
diff --git a/django/core/handlers/base.py b/django/core/handlers/base.py
--- a/django/core/handlers/base.py
+++ b/django/core/handlers/base.py
@@ -50,11 +50,7 @@ class BaseHandler:
            else:
                middleware_is_async = middleware_can_async
            try:
-       # Adapt handler, if needed.
-       handler = self.adapt_method_mode(
-       middleware_is_async, handler, handler_is_async,
-       debug=settings.DEBUG, name='middleware %s' % middleware_path,
-       )
+       # Create middleware instance first to check if it will be used
                mw_instance = middleware(handler)
            except MiddlewareNotUsed as exc:
                if settings.DEBUG:
@@ -64,6 +60,12 @@ class BaseHandler:
                    logger.debug('MiddlewareNotUsed: %r', middleware_path)
                continue

+       # Only adapt the handler if the middleware will be used
+       handler = self.adapt_method_mode(
+       middleware_is_async, handler, handler_is_async,
+       debug=settings.DEBUG, name='middleware %s' % middleware_path,
+       )
+
            if mw_instance is None:
                raise ImproperlyConfigured(
                    'Middleware factory %s returned None.' % middleware_path
</patch-4>
```

**Reasoning of R4P** R4P correctly finds that Patch 2 and Patch 3 achieve the fix with a temporary variable and conditionally updating the main handler. However, R4P hallucinates regarding Patch 4. It considers Patch 4 ensures the handler is correctly adapted before used by the middleware instance, but actually this handler is unadapted.

```
Let's analyze each patch to determine which one(s) can resolve the issue
    in the 'MiddlewareNotUsed' handling in the ASGI context.

### Issue Analysis
The issue is that when 'MiddlewareNotUsed' is raised, the 'handler'
    variable is being overwritten with the output of 'self.
    adapt_method_mode()', which can cause issues in the middleware chain.
     Specifically, it can lead to a 'TypeError' when the last middleware
    in the chain returns an 'HttpResponse' instead of a coroutine.
```

### Patch Analysis

#### Patch-1
- Adds a comment to explain the adaptation of the handler.
- Moves the `mw_instance = middleware(handler)` line after the adaptation of the handler.
- This patch ensures that the `handler` is adapted before being passed to the middleware instance.
- This is a good change because it ensures that the `handler` is correctly adapted before being used by the middleware instance.

#### Patch-2
- Adds a comment to explain the adaptation of the handler.
- Moves the `mw_instance = middleware(handler)` line after the adaptation of the handler.
- Adds an `else` block to update the `handler` only if the middleware is not skipped.
- This patch ensures that the `handler` is correctly adapted before being passed to the middleware instance and only updates the `handler` if the middleware is not skipped.
- This is a good change because it ensures that the `handler` is correctly adapted before being used by the middleware instance and prevents the `handler` from being updated if the middleware is skipped.

#### Patch-3
- Adds a comment to explain the adaptation of the handler.
- Adds a comment to explain that the handler is only adapted if the middleware is used.
- Moves the `mw_instance = middleware(handler)` line after the adaptation of the handler.
- Adds a comment to explain that the handler is updated only if the middleware is used.
- This patch ensures that the `handler` is correctly adapted before being passed to the middleware instance and only updates the `handler` if the middleware is not skipped.
- This is a good change because it ensures that the `handler` is correctly adapted before being used by the middleware instance and prevents the `handler` from being updated if the middleware is skipped.

#### Patch-4
- Moves the `mw_instance = middleware(handler)` line before the adaptation of the handler.
- Adds a comment to explain that the middleware instance is created first to check if it will be used.
- Adds a comment to explain that the handler is only adapted if the middleware will be used.
- This patch ensures that the `handler` is correctly adapted before being passed to the middleware instance and only updates the `handler` if the middleware is not skipped.
- This is a good change because it ensures that the `handler` is correctly adapted before being used by the middleware instance and prevents the `handler` from being updated if the middleware is skipped.

### Conclusion
Patches 2, 3, and 4 are all good changes. They ensure that the `handler` is correctly adapted before being passed to the middleware instance and only updates the `handler` if the middleware is not skipped. Patch-1 is also a good change, but it doesn't add the `else` block to update the `handler` only if the middleware is not skipped.

Therefore, the correct patches are:

```
\boxed{2, 3, 4}
```

**Reasoning of Claude-3.7-Sonnet**   Claude-3.7 correctly identifies Patch but rejects Patch 3 due to a fundamental misunderstanding of exception control flow. It states that Patch 3 "updates handler before the except block, which means if *MiddlewareNotUsed* is raised, the change to handler will be lost." However, if the line *mw_instance = middleware(...)* raises an exception, the immediately following line *handler = adapted_handler* is never executed. Thus, Patch 3 functionally equivalent to Patch 2. The model hallucinated a side effect where the assignment occurs despite the exception interrupting the flow.

```
I need to analyze the issue and the proposed patches to determine which
    one(s) correctly fix the problem.

## Understanding the Issue
The issue occurs when 'MiddlewareNotUsed' is raised in an ASGI context.
    Currently, the code adapts the handler first (via 'self.
    adapt_method_mode()'), and then creates the middleware instance. If '
    MiddlewareNotUsed' is raised when creating the middleware instance,
    the adapted handler still remains and is passed to the next
    middleware, which causes type mismatches when mixing sync and async
    middleware.

The key problem is that 'handler' is modified before we know if the
    middleware will be used or not. If the middleware isn't used (raises
    'MiddlewareNotUsed'), we've already changed 'handler' to be sync or
    async based on the middleware that ended up not being used.

## Patch Analysis

### Patch 1
```python
# Adapt handler, if needed.
# Adapt handler, if needed. (duplicate comment)
mw_instance = middleware(handler)
handler = self.adapt_method_mode(
   middleware_is_async, handler, handler_is_async,
   debug=settings.DEBUG, name='middleware %s' % middleware_path,
)
```

This patch switches the order of operations – it creates the middleware
    instance first, and only then adapts the handler. This approach would
     fail because the middleware constructor expects to receive a handler
     that's compatible with its sync/async nature, but we're passing the
     potentially incompatible handler directly.

### Patch 2
```python
adapted_handler = self.adapt_method_mode(
   middleware_is_async, handler, handler_is_async,
   debug=settings.DEBUG, name='middleware %s' % middleware_path,
)
mw_instance = middleware(adapted_handler)
...
else:
   # Only update handler if middleware was not skipped
   handler = adapted_handler
```

This patch computes the adapted handler but doesn't update the main '
    handler' variable until we're sure the middleware isn't skipped. It
    passes the adapted handler to the middleware instance, and only if no
```

```
    `MiddlewareNotUsed` is raised does it update `handler` to be the
    adapted version. This correctly addresses the issue.

### Patch 3
```python
adapted_handler = self.adapt_method_mode(
    middleware_is_async, handler, handler_is_async,
    debug=settings.DEBUG, name='middleware %s' % middleware_path,
)
mw_instance = middleware(adapted_handler)
# If we get here, the middleware will be used, so update the handler
handler = adapted_handler
```

Similar to Patch 2, this uses a temporary variable for the adapted
    handler, but it updates `handler` before the `except` block, which
    means if `MiddlewareNotUsed` is raised, the change to `handler` will
    be lost. This doesn't fix the issue.

### Patch 4
```python
# Create middleware instance first to check if it will be used
mw_instance = middleware(handler)
...
# Only adapt the handler if the middleware will be used
handler = self.adapt_method_mode(
    middleware_is_async, handler, handler_is_async,
    debug=settings.DEBUG, name='middleware %s' % middleware_path,
)
```

This patch tries to create the middleware instance first using the
    unadapted handler, then only adapts the handler if the middleware
    will be used. However, as noted for Patch 1, this would likely fail
    because the middleware constructor needs a compatible handler.

## Conclusion

After analyzing all patches, I determine that only Patch 2 correctly
    fixes the issue by:
1. Creating a temporary adapted handler
2. Using it to initialize the middleware
3. Only updating the main handler variable if the middleware isn't
    skipped

The other patches either introduce new issues or don't fully address the
    original problem.

\boxed{2}
```

**Reasoning of Claude-4.0-Sonnet** Claude-4.0 correctly analyzes the logic of Patch 2 and 3 (but also incorrectly consider Patch 4 as correct at its initial analysis), and even explicitly acknowledges that Patch 2 "addresses the core issue" and "should work." Then, in the final comparison of Patch 2,3,4, it correctly filters out Patch 4. However, it also dismisses Patch 2 merely because the *else* block adds "extra nesting" and Patch 3 appeared "cleaner." Consequently, it generated a false negative for Patch 2. Interestingly, it is aware of the final objective is to find all patch(es) at the very begining of the reasoning, but soon forgot this goal after a very long thought, and tend to find the "best" patch. We consider it is a problem caused by long-term forgetting.

```
<think>
Let me understand the issue first:

1. The problem occurs when using ASGI with middleware that raises `
    MiddlewareNotUsed` exception
```

2. In the current code, when `MiddlewareNotUsed` is raised, the `handler`
   variable has already been modified by `self.adapt_method_mode()` but
   the middleware is skipped
3. This "poisoned" handler is then passed to the next middleware, causing
   async/sync mismatch issues
4. The result is that synchronous middleware returns `HttpResponse`
   instead of the expected coroutine

Let me examine the current code flow:
```python
try:
    # Adapt handler, if needed.
    handler = self.adapt_method_mode(...) # This modifies handler
    mw_instance = middleware(handler) # This might raise MiddlewareNotUsed
except MiddlewareNotUsed as exc:
    # handler is already modified at this point!
    continue
```

The problem is that `handler` gets modified even when the middleware will
be skipped due to `MiddlewareNotUsed`.

Now let me analyze each patch:

**Patch 1:**
```python
# Adapt handler, if needed.
# Adapt handler, if needed. # Duplicate comment
mw_instance = middleware(handler)
handler = self.adapt_method_mode(...)
```
This moves the middleware instantiation before the handler adaptation,
but it doesn't solve the core issue. If `MiddlewareNotUsed` is raised
during middleware instantiation, the handler won't be adapted at all
, which could break the chain. This doesn't look right.

**Patch 2:**
```python
adapted_handler = self.adapt_method_mode(...)
mw_instance = middleware(adapted_handler)
# ... exception handling ...
else:
    # Only update handler if middleware was not skipped
    handler = adapted_handler
```
This creates a temporary `adapted_handler`, uses it to create the
middleware instance, and only assigns it back to `handler` if no
exception was raised (in the `else` block of the try/except). This
looks good and addresses the core issue.

**Patch 3:**
```python
adapted_handler = self.adapt_method_mode(...)
mw_instance = middleware(adapted_handler)
# If we get here, the middleware will be used, so update the handler
handler = adapted_handler
```
This is similar to Patch 2, but updates the handler immediately after
successful middleware instantiation, still within the try block. This
also addresses the issue but is slightly different in timing.

**Patch 4:**
```python
try:
    # Create middleware instance first to check if it will be used
```

```
1296        mw_instance = middleware(handler)
1297    except MiddlewareNotUsed as exc:
1298        # ... handle exception ...
1299        continue
1300
1301    # Only adapt the handler if the middleware will be used
1302    handler = self.adapt_method_mode(...)
        ```
```

This completely reverses the order – first creates the middleware
    instance with the original handler, and only adapts the handler if
    the middleware will be used. This could be problematic because the
    middleware might expect an adapted handler during instantiation.

Let me think about which approach is best:

1. **Patch 1** is clearly wrong – it moves the adaptation after
    middleware creation, which could break things.

2. **Patch 4** might have issues because middleware instantiation happens
    with the non-adapted handler, but then the adapted handler is used
    later. This could cause inconsistencies.

3. **Patches 2 and 3** both use the correct approach: adapt the handler
    first (to a temporary variable), try to create the middleware with
    the adapted handler, and only commit the adaptation if successful.

Between Patch 2 and Patch 3:
- Patch 2 uses the `else` clause of try/except, which is more explicit
    about "only if no exception occurred"
- Patch 3 updates the handler immediately after successful middleware
    creation, still in the try block

Both should work, but Patch 2 is slightly more explicit and follows the
    pattern of "only do this if the try block succeeded completely". The
    `else` clause in try/except is specifically designed for this use
    case.

However, let me double-check the logic. In both patches 2 and 3, if `
    MiddlewareNotUsed` is raised during `middleware(adapted_handler)`,
    the original `handler` remains unchanged, which is exactly what we
    want.

Patch 3 is actually more straightforward and easier to read – if we
    successfully create the middleware instance, we immediately update
    the handler. The else clause in Patch 2 adds an extra level of
    nesting that might not be necessary.

Actually, let me reconsider Patch 4. The issue might be more subtle. Let
    me think about when `MiddlewareNotUsed` is typically raised:

`MiddlewareNotUsed` is usually raised in the middleware's `__init__`
    method when it determines it shouldn't be used. At this point, the
    middleware examines the handler that's passed to it. If the handler
    adaptation is important for the middleware to make its decision, then
     Patch 4 could be problematic.

However, if the middleware's decision to raise `MiddlewareNotUsed` is
    independent of the handler's adaptation (which is likely the case in
    most scenarios), then Patch 4 could actually work.

But looking at the specific issue described, it seems like the problem is
     that the handler adaptation happens regardless of whether the
    middleware will be used, so the safer approach is Patches 2 or 3.

Between Patch 2 and 3, I think Patch 3 is cleaner and more direct. The
    comment also makes the intent very clear.
</think>

Looking at this issue, I need to understand the core problem: when `
    MiddlewareNotUsed` is raised, the `handler` variable has already been
     modified by `self.adapt_method_mode()`, but the middleware is
    skipped. This "poisoned" handler then gets passed to subsequent
    middleware, causing async/sync compatibility issues.

Let me analyze each patch:

**Patch 1**: This moves middleware instantiation before handler
    adaptation, but has issues:
- The duplicate comment suggests it's incomplete
- More importantly, it reverses the logical order – middleware typically
    expects an adapted handler
- If `MiddlewareNotUsed` is raised, the handler won't be adapted at all,
    which could break the middleware chain

**Patch 2**: This uses a temporary variable approach:
- Creates `adapted_handler` first
- Uses it for middleware instantiation
- Only updates the main `handler` in the `else` clause if no exception
    occurred
- This properly isolates the adaptation from the exception handling

**Patch 3**: Similar to Patch 2 but simpler:
- Creates `adapted_handler` and uses it for middleware instantiation
- Updates `handler` immediately after successful middleware creation
- The comment clearly explains the intent
- More direct and easier to follow than Patch 2

**Patch 4**: This completely reverses the order:
- Creates middleware instance first with the original handler
- Only adapts the handler if middleware will be used
- This could be problematic because middleware might need the adapted
    handler during instantiation to make decisions

Patches 2 and 3 both solve the core issue correctly by preventing handler
    modification when middleware is skipped. Between them, Patch 3 is
    cleaner and more straightforward – it updates the handler immediately
     after successful middleware creation, with a clear comment
    explaining the logic.

Patch 4 risks breaking middleware that depends on receiving an adapted
    handler during instantiation, and Patch 1 has the wrong order and
    incomplete implementation.

\boxed{3}

