# OpenReview forum: "Scalable Supervising Software Agents with Patch Reasoner"
_ICLR.cc/2026/Conference — Submitted to ICLR 2026_

### Official Review · Reviewer_dq9x · 2025-10-31

**Soundness:** 3
**Presentation:** 3
**Contribution:** 3
**Rating:** 6
**Confidence:** 4

**Summary:**

The paper studies an interesting problem of learning "execution free patch verifiers" for SWE agents. Currnent test time scaling agents fall into 3 categories: execution-free verifiers, execution-based verifiers and hybrid verifiers. Current execution-free verifiers require access to agent trajectory which can bias the final response of the patch verifier. The paper simply considers learning better "exec ution free patch verifiers" as a reasoning problem which can be solved with RL. The proposed approac R4P achieves 72% "verification accuracy" while also help improve TTS (test time scaling) performance by ~10% on SWE-Bench-Verified. (Its important to note that 72% is verification accuracy, not accuracy after TTS)

**Strengths:**

* The idea of using RL for training patch verifiers is interesting.

* "the binary outcome reward is very easy to hack, making the training unstable." This is a very insightful observation especially for training patch verification agents with RL.

* The final results show improvements for patch verification and TTS performance.

**Weaknesses:**

* The paper mentions that "The Pass@1 resolution rate on SWE-bench-verified steadily improves with more training data and finally reaches 26.2%, outperforming Lingma Agent +
Lingma SWE-GPT-72B (Ma et al., 2024)." Have the authors also tried their approach on closed source models like Claude 4.5 Sonnet and best open source models like R2E-Agent, SWE-Smith, DeepSWE, etc.?

* The paper considers group verification for training the RL model (sec. 3). Have the authors tried training a patch verifier using RL for individual patches given the input problem statement and repository sandbox?

* The authors mention that for training they use "2,438 issue instances". Have the authors explored impact of number of issues on the performance of the patch verifier? Is 2500 issues enough for RL training?


* For fig. 2, I believe r2e-gym also has an execution based verifier. I will be curious on how the proposed verifier compares to execution-based verifiers?

* Also for Tab. 1, are all values reported for execution-free verification given just the problem statement and final patch?

* Using RL for training the patch verifier is interesting. Can the authors please also share some outputs from the patch verifier to help understand how RL shapes the reasoning process as compared to closed-source patch verifiers like o3, gpt-5 etc?

* Finally, while not a major concern, have the authors tried comparing R4P with recent pretrained closed-source patch verifiers like Claude 4.5 Sonnet, gpt-5 etc?

**Questions:**

Please see the weaknesses section for some additional questions.

---

> ### Author Response · Authors · 2025-11-22
>
> **Q1**: *The paper mentions that "The Pass@1 resolution rate on SWE-bench-verified steadily improves with more training data and finally reaches 26.2%, outperforming Lingma Agent + Lingma SWE-GPT-72B (Ma et al., 2024)." Have the authors also tried their approach on closed source models like Claude 4.5 Sonnet and best open source models like R2E-Agent, SWE-Smith, DeepSWE, etc.?*
>
> **A1**: Thank you for the suggestion. We first want to clarify that Mini-SE is not the main focus of this paper. Our goal is to show that training LLM (Qwen) on a very light-weighted scaffold (Mini-SE) with R4P supervision can effectively improve LLM’s performance on SWE, thereby showing R4P’s practicality on scalable RL scenario. We intentionally make Mini-SE light-weighted so that it will not cost much resource and time for R4P’s practicality evaluation (i.e., RL training on Mini-SE).
>
> Furthermore, R2E-Agent, SWE-Smith (SWE-agent), and DeepSWE are all agentic scaffolds. Their underlying LLMs are also Qwen fine-tuned on their proposed datasets. While we also use Qwen as the base model for Mini-SE, there is a fundamental difference in purpose: We focus on validating whether R4P can provide a practical reward to support stable and scalable RL without causing training collapse, while they focus on whether their collected training dataset has better quality and can produce better SFT-trained models than other datasets. To achieve our goal, we need to perform RL training on Qwen and use the stable scaling reward curve to demonstrate R4P’s practicality, rather than continuing training or comparing/surpassing these existing agent’s fine-tuned models. In addition, we cannot train Claude 4.5 with R4P as it is closed-sourced.
>
> However, if reviewer is meant to understand R4P's effectiveness on simply verifying the patches generated by these advanced models (rather than validating RL practicality), then we include only models with offline patches (like o3 and DeepSWE) since their patches are labeled and accessible. We do not additionally sample patches for other models to evaluate R4P, as it results in extra cost while the accessible offline patches is already sufficiently large.
>
> **Q2**: *The paper considers group verification for training the RL model (sec. 3). Have the authors tried training a patch verifier using RL for individual patches given the input problem statement and repository sandbox?*
>
> **A2:** We did train and compare a patch verifier with RL on individual patches with only the input problem statement. This experiment is shown in Figure 7 (the point-wise verifier). As we discussed in Section 2, the binary output of a point-wise verifier is easy to hack, and unlike the group-wise setting, it lacks the mutual reference and rich context across patches. As a result, its performance is weaker and the training is much harder to stabilize.
>
> In addition, we want to clarify the “repository sandbox” is a runtime environment required for executing software, typically including many executable binaries such as third-party libraries or system libraries in the computer (e.g., .dll, .so, .lib, etc.). These environments do not contain any textual semantic information. In practice, a sandbox is usually a Docker container. Therefore, we could not feed sandbox environment into the model as input.
>
> However, if reviewer is meant to understand if it is possible to train a verifier to use the problem statement and patches while *“leveraging”* the sandbox to perform verification, then this would in fact be an *unscalable test-based agentic verifier*, as it still depends on building a sandbox (which is exactly the challenges that R4P aims to tackle in Section 2). Therefore, our main baselines are all execution-free verifiers.
>
> **Q3:** *The authors mention that for training they use "2,438 issue instances". Have the authors explored impact of number of issues on the performance of the patch verifier? Is 2500 issues enough for RL training?*
>
> **A3:** Thank you for the question. In our early-staged demo experiment, we actually found that R4P’s performance quickly saturates as the issues increases: When using only half of the issue instances (1.2k) for training, R4P still achieves a verification accuracy of 70.2, which is only 2% lower than training with the full 2.4k issues.
>
> In addition, most existing training datasets for SWE agents contain only around 2k real-life issues, so in practice it is difficult to obtain more for training. However, it should be note that R4P is a verification model. Its input should be a issue-patch pair. Thus, while we only have 2.4k issue, each issue is actually paired with 6 sampled patches, resulting in a 6 * 2.4k issue-patch pairs as training input. It is sufficient for RL, as the model’s “answer” in RL are dynamically sampled (rollout) throughout RL training process. As a result, the total number of (issue-patch, prediction) input–output pairs becomes very large.
>
> Thus, we believe the dataset is enough for training R4P.

---

> ### Author Response · Authors · 2025-11-22
>
> **Q4:** *For fig. 2, I believe r2e-gym also has an execution based verifier. I will be curious on how the proposed verifier compares to execution-based verifiers?*
>
> **A4:** Thank you for the question. Our main experiments (Table 1 and Figure 2) focus on comparing R4P against existing *scalable* *supervision* methods. For this reason, we do not include unscalable baselines that require building execution environments, including execution-based verifier like R2E-Test and DeepSWE-Test (which is actually a test generation&execution agent system).
>
> In addition, Figure 2 evaluates verifier performance on *offline* patches submitted in SWE-bench experiments. These patches themselves do not come with executable environments. It would require substantial engineering effort for integrating & debugging all patches generated by diverse LLM agents with each different execution-based verifier.
>
> Notably, our use of DeepSWE-Test in Figure 3 serves a different purpose: it provides a baseline for the TTS setting to illustrate R4P’s TTS practicality generalization across scaffolds (DeepSWE). This comparison is feasible only because DeepSWE has already released the predictions of their execution-based verifier (DeepSWE-Test) on DeepSWE agent–generated patches. In contrast, R2E’s execution-based verifier does not provide such *ready-to-use* results. Therefore, we do not include execution-based verifiers such as DeepSWE-Test or R2E’s test agent in the main experiment (Figure 2).
>
> **Q5:** *Also for Tab. 1, are all values reported for execution-free verification given just the problem statement and final patch?*
>
> **A5:** All results in Table 1 use the same group-wise setup as R4P, i.e., each verifier sees the problem statement and a group of diverse patches. As shown in Figure 5 and 6, this formulation allows every baseline to achieve its highest possible accuracy, ensuring a fair comparison.
>
> **Q6:** *Using RL for training the patch verifier is interesting. Can the authors please also share some outputs from the patch verifier to help understand how RL shapes the reasoning process as compared to closed-source patch verifiers like o3, gpt-5 etc?*
>
> **A6:** We thank the reviewer for the suggestion. We performed a detailed case study on the reasoning trajectory of R4P, as well as the output of closed-source models (e.g., Claude-3.5-Sonnet and Claude-4.0-Sonnet) in the Appendix C for reference. Note that the reasoning contents of the reviewer recommended closed-source models (o3 and GPT-5) are inaccessible: their outputs contain only the final answers without reasoning (e.g., output `\boxed{1, 4}` only). Thus, we have to use hybrid-reasoning models (whose reasoning processes are visible) as substitutes. Due to the space limit (the reasoning trajectories are rather long), we will not place them here on OpenReview.
>
> **Q7:** *Finally, while not a major concern, have the authors tried comparing R4P with recent pretrained closed-source patch verifiers like Claude 4.5 Sonnet, gpt-5 etc?*
>
> **A7:** We thank the reviewer for the suggestion. Although these models *had not yet been released by our main experiment of R4P*, we still believe that comparing against the strongest available models is meaningful for future references. The results are as follows:
>
> | Model | Acc. | F1 | EM |
> | --- | --- | --- | --- |
> | Claude-4.5-Sonnet | 71.6 | 55.0 | 30.4 |
> | GPT-5 | **76.2** | 62.4 | 41.5 |
> | R4P | 72.2 | **63.3** | **41.8** |
>
> The results demonstrate that even with a significantly smaller parameter size (32B) and an older base model (Qwen2.5), *R4P still maintains advantages in terms of F1 and EM*.
>
> Note that there are also many advanced open-source models (e.g., Qwen3-Coder MoE) released recently. Since R4P's base model (Qwen2.5) and our original closed-source baseline (e.g., o3) belong to a previous era of LLM, we believe a direct comparison with GPT-5 and Claude-4.5-Sonnet (rather than o3 or Claude-4.0-Sonnet) is *unfair to R4P*. We release these data points primarily to serve as a reference for future work that aims to train stronger open-source patch reasoner models on more advanced foundational backbones.
>
> We hope these results help alleviate your concerns.

---

### Official Review · Reviewer_N1rv · 2025-10-31

**Soundness:** 2
**Presentation:** 3
**Contribution:** 3
**Rating:** 8
**Confidence:** 4

**Summary:**

The paper introduces R4P (Reasoning-for-Patch) — a scalable, test-free reward model for supervising software engineering (SWE) agents via reasoning rather than traditional test execution.
It addresses the scalability bottleneck of test-based verification, which is heavy, fragile, and limited by test coverage.
R4P formulates patch verification as a group-wise reasoning task, comparing multiple candidate patches to produce dense, stable rewards during reinforcement learning (RL).
Experiments show that R4P achieves 72.2% patch verification accuracy, surpassing proprietary models like OpenAI o3. The authors further train a lightweight agent, Mini-SE, purely under R4P supervision, which achieves 26.2% Pass@1 on SWE-bench-verified—+10% over Qwen3-32B—and 33.8% when combined with R4P at test time.
The paper argues that R4P enables scalable supervision for SWE agents without dependence on sandbox testing

**Strengths:**

- The group-wise reasoning objective transforms sparse binary verification into a dense, stable reward signal
- The paper provides ample evidence showing the advantage of R4P, along with nice ablation studies to analyze the behavior of R4P model

**Weaknesses:**

- The reward model is fixed post-training, leading to potential reward drift as agents improve. It will be interesting to understand the RL behavior when you overtrain the model with such a static reward model model

- In Fig. 9, it will be good to draw the confidence interval to see if the trend is significant. The bins to the right have too few samples, which makes the conclusion that "verification accuracy positively correlates with/ number of edited lines" a bit ungrounded

- Despite the two challenges of applying R4P directly to existing agent scaffolds via RL, it'd be interesting to demonstrate R4P's ability to provide supervision for training models to work on general agent scaffolds. e.g., you can use R4P to re-rank patches/trajectories generated on training datasets like SWE-Gym+OpenHands and SFT on the top 10% trajectories and measure performance improvements on OpenHands vs. random sampling. This could demonstrate R4P's ability to generalize across scaffolds.

**Questions:**

> As the agent’s policy improves, the static reward model may become misaligned with true answer quality
I wonder whether the authors have tried to overtrain the policy (i.e., training the model longer in Figure 3a). I'd be interested in understanding the R4P approach's bottleneck, e.g., whether it will saturate at a fixed performance on SWE-Bench or degrade performance if overtrained.

- I would be helpful if the authors could share more details about how the Acc/F1/EM was calculated, as well as the exact reward function that was used to perform RL on Mini-SE LM.

---

> ### Author Response · Authors · 2025-11-22
>
> **Q1**: *The reward model is fixed post-training, leading to potential reward drift as agents improve. It will be interesting to understand the RL behavior when you overtrain the model with such a static reward model model. - As the agent’s policy improves, the static reward model may become misaligned with true answer quality I wonder whether the authors have tried to overtrain the policy (i.e., training the model longer in Figure 3a). I'd be interested in understanding the R4P approach's bottleneck, e.g., whether it will saturate at a fixed performance on SWE-Bench or degrade performance if overtrained.*
>
> **A1**:  Thank you for the insightful question. The impact of RM’s static feature on SWE agent training is also a concern we have been paying attention to. As discussed in the Limitations section, when using a static RM, as the policy $\pi$ of the trained model gradually deviates from the initial policy $\pi_{init}$ (i.e., $D_{KL}(\pi||\pi_{init})$ increases), and the gap between the proxy RM’s score and the golden reward grows. This can lead to ineffective or even harmful training, e.g., the proxy reward increasing while the golden reward saturates or even decreases.
>
> During training Mini-SE with R4P, we observed a similar phenomenon: While the local training reward was still slowly increasing, the test accuracy dropped sharply after stabilizing for some time. At step 300, the test accuracy fell from 24.4 (step 290) to 21.6. We consider it is a signal of overtrain due to (1) the test accuracy are relatively stable for a certain period before this drop, (2) the magnitude of this drop was much larger than before (e.g., from 22.8 to 20.8 between steps 170 and 180), and (3) the training has gone through 4 full epochs. Since R4P is essentially a static reward model, we treat this as reaching the limit of R4P training (and we indeed expect it to saturate as some specific step). Given that Agentic RL is costy and slow, we stopped training at this point and did not explore further.
>
> Therefore, we consider R4P has the inherent limitations of static reward models. At least under overtraining (4 epochs, 300 steps), there is no significant gain, and misaligned rewards can lead to degraded performance. In the Limitations section, we discuss the potential use of staged reward policy alignment to mitigate this problem. Moreover, given the limited amount of training data, overfitting (4 epochs) can naturally lead to saturation or performance degradation. In scaling scenarios with larger data, we expect that such saturation or degradation would occur at later steps.
>
> **Q2**: *In Fig. 9, it will be good to draw the confidence interval to see if the trend is significant. The bins to the right have too few samples, which makes the conclusion that "verification accuracy positively correlates with/ number of edited lines" a bit ungrounded*
>
> **A2**: Thank you for the suggestion. We have redrawn Figure 9 and updated it in the paper. Specifically, we added the Wilson score confidence interval for each bin (95% confidence level) and the number of correct patches / total patches. Notably, a total of 1,340 patches were evaluated in our experiments, and the smallest bin still contains 44 samples. This number is larger than some commonly-used datasets such as AIME24/25 (30 samples for each). Thus, we consider the sample number in each bin is not that little.
>
> In the revised figure, verification accuracy and the number of edited lines still generally show a positive correlation. However, we also observed that the lower bound of the confidence interval in the 26–50 range is slightly lower than in the 1–25 range. This suggests that our previous conclusion may be too absolute. We therefore soften the statement in the paper as follows:
>
> > As shown in Fig. 9, R4P's verification accuracy generally appears to be positively correlated with the number of edited lines in a patch. This trend suggests that R4P's cross-referencing of a patch's edition content and position as context contributes to effective patch verification.

---

> > ### Author Response · Authors · 2025-11-22
> >
> > **Q3**: *Despite the two challenges of applying R4P directly to existing agent scaffolds via RL, it'd be interesting to demonstrate R4P's ability to provide supervision for training models to work on general agent scaffolds. e.g., you can use R4P to re-rank patches/trajectories generated on training datasets like SWE-Gym+OpenHands and SFT on the top 10% trajectories and measure performance improvements on OpenHands vs. random sampling. This could demonstrate R4P's ability to generalize across scaffolds.*
> >
> > **A3**:  We thank the reviewer for the insightful suggestion. We agree that using R4P for rejection sampling & SFT on general SWE’s agents is indeed a very interesting angle for validating cross-scaffold generalization ability. However, we also believe that even if we conducted this on trajectories of SWE-Gym + OpenHands, the results would not illustrate R4P’s generalization ability. This is because the only available data collection pipeline we have currently is the SWE-Gym + OpenHands pipeline, which has *already been used for R4P’s RL training* (Section 3). If we were to use R4P again to perform rejection sampling on this same data and then train OpenHands, this would lead to data contamination (i.e., “testing on the training set”) since R4P has already seen these patches and explicitly optimized to find good ones. Therefore, even if we conducted the experiment, the results would likely not be *generalizable*.
> >
> > Additionally, OpenHands trajectories are recorded in a Fibonacci-style scheme. When the number of steps is large, the retained trajectories result in an extremely large number of fragment files, consuming significant disk space and file handles. Since R4P training only requires patches, we actually preserved only the patches in our earlier experiments.
> >
> > However, we do agree that rejection sampling + SFT is indeed another considerable downstream task (beyond RL and TTS) for evaluating verifier’s practicality. To evaluate it, we would need to deploy a new data collection pipeline and setup & sampling with a new executable data environment + agent framework (e.g., SWE-Smith + SWE-Agent). This would require substantial effort, API budget, and time. However, in our future work for addressing RM’s static limitation, we can consider evaluating it as a supplementary: Since solving the policy-alignment problem of static RM requires periodically aligning the reward with executable environment *different from the verifier-training data*, we will in any case need an environment distinct from the one used to train the verifier. Therefore, such environment can naturally serve as the basis for studying rejection-sampling + SFT generalization on general agents.
> >
> > Again, we sincerely thank the reviewer for the insightful suggestions. They offer excellent guidance for our future work.

---

> ### Author Response · Authors · 2025-11-22
>
> **Q4**: *I would be helpful if the authors could share more details about how the Acc/F1/EM was calculated, as well as the exact reward function that was used to perform RL on Mini-SE LM.*
>
> **A4**: We thank the reviewer for this suggestion. We will detail the exact definitions of the metrics utilized in our evaluation. (The following part is also added in the Appendix B in our revised manuscript.)
>
> If a dataset has $N$ patches and $M$ unique issues (where each issue corresponds to $K$ patches), then these $N$ patches can be grouped into $M$ patch groups (each of size $K$, i.e., $N = M \times K$). Below are the detailed definitions:
>
> **1. Acc**
>
> Accuracy is a patch-level evaluation metric that quantifies the proportion of correctly verified patches out of the total number. Let $y_i$ be the ground-truth label for patch $p_i$, and $y^*(p_i)$ be the model's prediction of its correctness. Using the indicator function $\mathbb{I}$ (which evaluates to 1 if the condition is true, and 0 otherwise), the accuracy is calculated as:
>
> $$
> Acc=\frac{\sum^N_{i=1}\mathbb{I}_{y_i = y^*(p_i)}}{N}
> $$
>
> **2. F1**
>
> The F1 score is a group-level evaluation metric. In our implementation, we calculate it using the set-theoretic definition (i.e., the Dice coefficient). For a given group $i$:
>
> - Let $A_i$ be the set of truly correct patches.
> - Let $B_i$ be the set of patches predicted as correct by the model.
>
> The F1 score for group $i$ is calculated as:
>
> $$
> F1_i=\frac{2|A_i\cap B_i|}{|A_i|+|B_i|}
> $$
>
> This metric serves to measure the similarity between the predicted set ($B_i$) and the ground-truth set ($A_i$). Furthermore, it can provide an alternative view of a retrieval task, i.e., the task of "retrieving the correct patches from a set of candidate patches." In this context, the F1 score represents the harmonic mean of Precision ($P_i$) and Recall ($R_i$):
>
> $$
> F1_i=\frac{2P_iR_i}{P_i+R_i}=\frac{2TP_i}{2TP_i+FP_i+FN_i}=\frac{2|A_i\cap B_i|}{|A_i|+|B_i|}
> $$
>
> Note that in our implementation, we calculate the overall average F1 score for each individual group:
>
> $$
> F1=\frac{\sum^M_{i=1}F1_i}{M}
> $$
>
> To handle the boundary conditions where one or both sets might be empty, we consider:
>
> - $F1_i = 1$ if both sets are empty, as the model correctly predicted that no patch is correct.
> - $F1_i = 0$ if only one set is empty, as it is a completely incorrect prediction.
>
> **3. EM**
>
> Exact Match is a group-level metric that assesses the model's ability to correctly verify all patches within an entire group simultaneously. It is formally defined as the proportion of patch groups where the predicted set of correct patches exactly matches the ground-truth set:
>
> $$
> EM=\frac{\sum^M_{i=1}\mathbb{I}_{|A_i|=|B_i|}}{M}
> $$
>
> **4. Agent Reward**
>
> For the RL objective on the Mini-SE LM, we employed the most common and straightforward reward function:
>
> $$
> Reward = 1 \text{ if patch is \`\`correct", else } 0 \text{ if patch is \`\`incorrect"}
> $$
>
> where the “correctness” of the patch is determined by the R4P.

---

### Official Review · Reviewer_sfSF · 2025-11-01

**Soundness:** 3
**Presentation:** 3
**Contribution:** 3
**Rating:** 6
**Confidence:** 3

**Summary:**

This work proposes a reasoning-based framework for supervising large language model (LLM) agents in software engineering tasks without relying on computationally expensive or fragile testing environments.
To overcome these challenges, the authors introduce R4P, a reasoning reward model that performs group-wise patch verification—evaluating multiple code patches for a given software issue to determine correctness via reasoning rather than execution. This design produces dense, stable rewards and mitigates reward hacking. Built upon Qwen2.5-Coder-32B-Instruct, R4P achieves 72.2% accuracy on the SWE-bench-verified dataset, outperforming OpenAI’s o3 model.

**Strengths:**

1. Innovative Test-Free Supervision Paradigm.
R4P redefines software agent supervision as a reasoning task, eliminating the dependency on sandbox testing. This shift addresses scalability, cost, and fragility in existing solutions.
2. The reward model design is novel.

**Weaknesses:**

1. Scope of this work can be better elaborated.
2. The evaluation can be more comprehensive.

My major concen of this work is clarity and evaluation, I believe these shortcomings can be overcame before submitting the camera-ready version.

Why the reward design is technically sound and how it affects the learning? I think this is important when designing a reward function for reinforcement learning, and maybe it is better to elaborate that it is aligned with your objective to avoid reward hacking.
Equation (3) can be better explained, and similar problem happened in other places, please define and explain each symbol carefully.

Maybe it is better to include some real data if possible, the current tests are completely on sythetic data.
I'm curious about the results if the correctness ratio is imbalanced, seems this is more natural in real-world problems.

**Questions:**

N/A

---

> ### Author Response · Authors · 2025-11-22
>
> **Q1**: *Why the reward design is technically sound and how it affects the learning? I think this is important when designing a reward function for reinforcement learning, and maybe it is better to elaborate that it is aligned with your objective to avoid reward hacking.*
>
> **A1:** We would like to elaborate on the design philosophy of our group-wise reward. As discussed in Section 2, for each patch to be verified, the verifier output has only two possible outcomes: Pass or Fail. This binary output space is too sparse. Even if the verifier randomly guesses on patches that it cannot reliably judge, it still has a 50% chance of being correct. This leads the model to verify more patches correctly than it actually should, causing an overestimation problem (the False Positive Rate is much higher than the False Negative Rate, as shown in Figure 6), and the model becomes difficult to train to convergence (Figure 7).
>
> To address this issue, we need a reward that is harder to guess. As discussed in Section 3, we adopt a group-wise output formulation, requiring the model to verify multiple patches at once. This gives a denser output space ($2^N$ output candidates rather than $2$), and enables a denser reward function based on the verification accuracy within the group. The intuition is that: e*ven if the verifier has the same overestimation on each patch, a reasoning CoT that correctly verifies more patches in a group should be considered better than one that verifies fewer correctly.* Therefore, the reward should reflect this ordering, and the normalized group-wise verification accuracy is naturally a good metric for showing how much better one reasoning result is than another.
>
> We also revised our manuscript in Section 3 (”Reward Modeling”) as follows:
>
> > **Reward modeling**: To avoid reward hacking caused by the model’s random guesses on a patch’s binary correctness when uncertain, we adopt a continuous group-wise reward requiring the model to verify as many correct patches as possible (i.e., group-wise accuracy). The intuition is: a reasoning that correctly verifies more patches should be considered better than one that verifies fewer, even though certain random guessing is inevitable. Thus, the reward clearly reflect how much better one reasoning result is compared to another, rather than treating real correct reasoning as equivalent to a randomly guessed correct answer.
>
> **Q2:** *Equation (3) can be better explained, and similar problem happened in other places, please define and explain each symbol carefully.*
>
> **A2**: Thank you for pointing this out. We have revised Section 3 and Equation 3. The revised explanation is also provided below:
>
> > … Furthermore, this group-wise approach transforms the binary outcome space into a much denser one (e.g., $\sum C_N^i$ possibilities for random selecting correct patches from a group $P=[p_1,...,p_N]$), which provides a richer supervision signal and significantly mitigates the reward hacking risk inherent in simple binary classification tasks. Specifically, during RL training, the reward of R4P $r(P)$ is calculated by the indicator function $\mathbb{I}_{y_i = y^\*(p_i)}$, which equals 1 if the predicted label $y_i$ for patch $p_i$ matches the ground-truth label $y^\*(p_i)$, and 0 otherwise.
>
> We also noticed that Equation 2 contains unexplained indicator function $\mathbb{I}_{y_i = y^*(p_i)}$, so we revised the text as well. The updated content is as follows:
>
> > … TTS aims to maximize the expectation $E$ of the number of model output $y^\*(x)$ equals to the ground-truth $y$, which is notated as indicator function $\mathbb{I}_{y=y^\*(x)}$ for a given prompt $x$, which equals 1 if the predicted label $y_i$ for patch $p_i$ matches the ground-truth label $y^\*(p_i)$, and 0 otherwise.

---

> > ### Author Response · Authors · 2025-11-22
> >
> > **Q3:** *Maybe it is better to include some real data if possible, the current tests are completely on synthetic data. I'm curious about the results if the correctness ratio is imbalanced, seems this is more natural in real-world problems.*
> >
> > **A3:** Thank you for the suggestion. Using patches from real-world PRs for evaluation is indeed much closer to real code review scenarios. However, R4P is a verifier designed for scalable supervising LLM agents on SWE task rather than reviewing human PRs on GitHub. Therefore, the data we sample from is aligned with the actual target of the verifier. In addition, in-the-wild patches are almost all accepted patches (i.e.,, “far more correct patches than incorrect patches”). This extreme imbalance make it very challenging to evaluate verifier’s ability to detect incorrect patches. Thus, we use agent-generated patches for evaluation.
> >
> > Even so, we value the reviewer’s concern about whether data imbalance may affect the verifier performance. Therefore, we present R4P’s verification accuracy across all agent’s patch subsest in our main experiment (Table 1) as a reference:
> >
> > | LLM Agents | Verification Accuracy | Ground-truth T : F |
> > | --- | --- | --- |
> > | 20250520-openhands-devstral-small | 74.03 | 66.6% : 33.4% |
> > | 20241029-openhands-codeact-2.1-sonnet-2024102 | 72.83 | 59.4% : 40.6% |
> > | 20240728-sweagent-gpt4o | 75.22 | 33.1% : 66.9% |
> > | 20240620-sweagent-claude3.5sonnet | 66.87 | 40.6% : 59.4% |
> > | overall | 72.23 | 49.9% : 50.1% |
> >
> > It is clear that though the overall dataset has a balanced label distribution, it is not label balanced in each individual LLM agent’s subset (whose correctness rate ranges from 30% to 70%, correlated with agent’s capability). The accuracy fluctuation generally within 6%, and does not correlate with different label distribution. Therefore, we consider that R4P is stable under data imbalance.

---

### Official Review · Reviewer_ueZU · 2025-11-02

**Soundness:** 3
**Presentation:** 3
**Contribution:** 3
**Rating:** 6
**Confidence:** 5

**Summary:**

This paper proposes to automatically evaluate software repository patches proposed by SWE agents via a verifier model that reasons explicitly about the proposed patches. They propose this as an explicit alternative to simple reward modeling that provides a sparse scalar estimate of quality of proposed patches. The proposed model, R4P, can be used to evaluate SWE agents and to train them in cases where a repository doesn't have ground-truth test cases. Experiments focus on evaluating R4P's quality as a verifier, and on the application of R4P for training SWE agents via RL and improving performance via test-time scaling. As far as I can tell, the main contribution is that R4P is trained to not only provide judgments over patch quality, but to also provide reasoning about its judgments. Augmenting models with reasoning capabilities has been a popular approach for improving model outcomes in difficult tasks, so it makes sense that it works well here too.

**Strengths:**

* The analysis in 5.3 is really comprehensive
* Experiments prove the efficacy of R4P on a simple agent scaffold, and strong performance as a verifier when compared with existing non-reasoning verifiers

**Weaknesses:**

* Figure 3 shows rewards for test data. Test data should be used in experiments very sparingly, to avoid compromising integrity of conclusions about model generalization
* Evaluation is only performed on the mini-SE scaffold, rather than other scaffolds that achieve stronger base performance on SWE-bench-verified. Would R4P still be useful in these other scaffolds? Do these scaffolds make R4P more difficult because they include much longer trajectories, which are more difficult to reason about (although R4P is trained on OpenHands trajectories, so it should be in-distribution to apply it to the OpenHands scaffold)?

**Questions:**

* What is the difference between DeepSWE-Verifier and DeepSWE-Test?

---

> ### Author Response · Authors · 2025-11-22
>
> **Q1**: *Figure 3 shows rewards for test data. Test data should be used in experiments very sparingly, to avoid compromising integrity of conclusions about model generalization*
>
> **A1:** Thank you for pointing it out. We found that it might be a misunderstanding due to our previously ambiguous Figure 3 caption: “*Mini-SE’s training and testing rewards during RL training*”. In fact, we did not use test data or test accuracy as the reward for model training. The test accuracy reported in Figure 3 are only the offline evaluation results of the cached model checkpoints, as we want to show that the test accuracy exhibits a *clear scaling curve* as the amount of training data increases, thereby showing the practicality of R4P on scalable RL training.
>
> To avoid misunderstanding, we have revised the caption, as shown below:
>
> > Figure 3: Left: Mini-SE’s training rewards and testing accuracy in RL process.
> >
>
> **Q2**: *Evaluation is only performed on the mini-SE scaffold, rather than other scaffolds that achieve stronger base performance on SWE-bench-verified. Would R4P still be useful in these other scaffolds? Do these scaffolds make R4P more difficult because they include much longer trajectories, which are more difficult to reason about (although R4P is trained on OpenHands trajectories, so it should be in-distribution to apply it to the OpenHands scaffold)?*
>
> **A2**: Thank you for the suggestion. We would like to clarify that:
>
> 1. The main evaluation (Table 1 and Figure 2) is performed on multiple agent scaffolds’ offline patches (and trajectories for those trajectory-based verifier baselines) on *SWE-Bench Experiments* (including SWE-Agent and OpenHands, as discussed in Section 5.1 and Appendix B.3). In fact, Mini-SE is used only to evaluate the *practicality* of R4P, i.e., whether R4P’s high-accuracy verification signal can truly support the downstream task of scalable RL and TTS (Figure 3). We intentionally makes Mini-SE to be very lightweight to reduce the computational cost for such costly RL downstream task evaluation.
> 2. R4P does not use trajectories for verification. It performs group-wise reasoning purely over patches. As we claim in Section 1, training a verifier that relies on specific agent’s trajectory (and specific interaction pattern) leads to limited generalization across different agents. This is also reflected in Figure 2, as their verification accuracy is far below our trajectory-free verifier (R4P). Therefore, R4P’s performance is not tied to the interaction pattern of any specific agent scaffold (e.g., OpenHands).
>
> Meanwhile, we appreciate the reviewer’s concern about analysis on in-distribution/out of distribution agent’s data. Thus, we have broken down R4P’s verification accuracy across different scaffolds and report the results below:
>
> | LLM Agents Experiments | Verification Accuracy | In-distribution Scaffold? |
> | --- | --- | --- |
> | 20250520-openhands-devstral-small | 74.03 | Yes |
> | 20241029-openhands-codeact-2.1-sonnet-2024102 | 72.83 | Yes |
> | 20240728-sweagent-gpt4o | 75.22 | No |
> | 20240620-sweagent-claude3.5sonnet | 66.87 | No |
> | overall | 72.23 | N/A |
>
> As shown, R4P’s performance is not significantly affected by whether the scaffold is OpenHands or not, as the best and the worst results are all illustrated on SWE-Agent’s patches.
>
> **Q3**: *What is the difference between DeepSWE-Verifier and DeepSWE-Test?*
>
> **A3**: DeepSWE-Verifier is a trajectory-based, execution-free verifier model trained on the DeepSWE agent‘s offline trajectory data. DeepSWE-Test is a test-based agent that generates and executes regression tests to evaluate whether the DeepSWE agent’s patches can pass them. The former is our main comparison target because it is scalable, as it does not require an executable environment. The latter depends on an individual executable environment for each issue, so it is not scalable and is not our main comparison target. We include DeepSWE-Test only to provide a baseline in a downstream TTS setting evaluation (as shown in Figure 3) to show R4P’s out-of-distribution generalization on cross-scaffold (DeepSWE) offline patch data.

---

### Author Response · Authors · 2025-11-27
**General Response**

We sincerely appreciate all reviewers for their supportive and positive feedback on our paper. We summarize our responses to common questions/suggestions before the end of rebuttal:

1. *Evaluation on other SWE agents* [ueZU, dq9x]: Our main experiment does evaluate R4P on *four different LLM + Agent combinations* (Appendix B.3) offline patches. Mini-SE is only used for validating R4P’s practicality on those costly downstream tasks: RL and TTS.
2. *Comparison with other Test agents* [ueZU, dq9x]: Our goal is to provide a *scalable supervision*. We do not include test agents in main experiment as they rely on pre-built sandboxes (Sec. 2). Yet, we do use DeepSWE-Test's offline results as it is available, enabling an extra support of R4P's generalizability on TTS without costly reproduction process.
3. *Ambiguity of symbols and metrics* [sfSF, N1rv]: We have revised our manuscript to clarify the symbols, metrics, and training rewards in both main text and appendix.

We hope these clarifications address your concerns. Thank you once again for your attention, support, and dedication.

---

### Meta-Review · Area_Chair_BZaR · 2026-01-04

**Summary:**

Important Note: while this wasn't discussed in reviews, I would rather reject this paper due to (serious) inconsistencies in math notation. For example, equation (2) does not make sense. Specifically, it does an argmax over the model parameters (in a section about test-time scaling) and involves an expectation over y∼D(θ|N,x), which does not make sense (we do not normally have a Bayesian distribution over model params in a conventional LLM, only a point estimate). Also, y is sampled from a distribution that does not involve y. I can only speculate the authors were trying to define some sort of 0-1 loss, but this is not the way to do it. Note that these are not typos, but strong indications of bigger underlying issues.

A second issue with the paper is the reliance on an indicator function in the definition of the reward in equation (3). It is not clear to me how labels indicating whether a software patch is correct without either manual inspection (expensive) or asking another LLM (biased). This one is not a deal-breaker, but the paper does not make the resolution of this problem sufficiently clear.

All reviews were superficial (while being marginally above the minimum bar for reviewers). The main issues they identified were:
1. Concerns about validity of evaluation (sufficient breath of the test set / initial concerns about test data leakage).
2. Concerns about clarity (relating to scope).
3. Requests for confidence bars.
4. Requests for more baseline models.
5. Concerns about the small size of the training set.

**Reviewer Concerns:**

1. Concerns about validity of evaluation (sufficient breath of the test set / initial concerns about test data leakage). These were resolved.
2. Concerns about clarity (relating to scope). (in my opinion unresolved)
3. Requests for confidence bars (resolved)
4. Requests for more baseline models (resolved to an extent).
5. Concerns about the small size of the training set (resolved)

**Reviewer Scores:**

I think the reviewers would leave their scores unchanged.

I don't think anyone would have reduced their rating (no critical issues were identified by reviewers, although the exist - see "summary")

The most enthusiastic reviewer already gave the paper an 8, and I doubt thy would move it up (this is definitely not a super-outstanding paper).

All reviewers gave superficial comments that have the appearance of being addressed on a lowest-effort basis, so I wouldn't expect score movement there either.

---

> ### Public Comment · ~Junjielong_Xu2 · 2026-03-04
> **Clarification on the Meta-Review**
>
> We thank AC's time in reviewing our submission and would like to respectfully clarify two points raised in the meta-review, as we believe they may stem from misunderstandings of our paper.
>
> 1. **Equation (2)**: θ is the **strategy of a test-time scaling approach** (e.g., the choice of verifier, number of rollouts, etc.), *not* the frozen LLM being tested. Eq.2 shows how an ideal test-time scaling strategy θ* should be (Line 104). *It does not involve any training or optimization process* on “argmax parameters”. The distribution of the final answer y involves x and θ because the test-time methods will filter or select a few candidate output samples after rollout (Line 107).
>
>     In addition, test-time formulation is a **background** rather than our contribution. It is proposed by DeepMind (Snell et al., 2024) (Line 103).
>
> 2. **Equation (3)**: Patch resolution correctness is verified via SWE-Gym (Line 204). Its **unit-test-based verification** is standard in the SWE research domain as discussed through out the abstract and introduction. The indicator in Equation (3) is a standard exact match indicator operator, which is also explained in Line 185 and rebuttal to Reviewer N1rv.
>
> We thank all four reviewers for their valuable feedback, which led to concrete improvements in our revision. We respectfully disagree with the characterization that the reviews were superficial.
>
> We hope these clarifications are helpful to readers and the community.

---

### Decision · Program_Chairs · 2026-01-26

Reject